

# Reconstruction of high-frequency methane atmospheric concentration peaks from measurements using metal oxide low-cost sensors

Rivera Martinez Rodrigo[1], Santaren Diego[1], Laurent Olivier[1], Broquet Gregoire[1], Cropley Ford[1], Mallet Cécile[2], Ramonet Michel[1], Shah Adil[1], Rivier Leonard[1], Bouchet Caroline[3], Juery Catherine[4], Duclaux Olivier[4], and Ciais Philippe[1]

[1]Laboratoire des Sciences du Climat et de l'Environnement, LSCE/IPSL, CEA-CNRS-UVSQ, Université Paris-Saclay, 91191 Gif-sur-Yvette, France
[2]Université de Versailles Saint-Quentin, UMR8190 – CNRS/INSU, LATMOS-IPSL, Laboratoire Atmosphères Milieux, Observations Spatiales, Quartier des Garennes, 11 Boulevard d'Alembert, 78280 Guyancourt, France
[3]SUEZ - Smart & Environmental Solutions; Tour CB21, 16 place de l'Iris, 92040 La Défense France
[4]TOTAL Energies - Raffinage chimie, Laboratoire Qualité de l'Air – 69360 SOLAIZE FRANCE

**Correspondence:** Rodrigo Rivera (rodrigo.rivera@lsce.ipsl.fr)

**Abstract.** Detecting and quantifying $CH_4$ gas emissions at industrial facilities is important goal for being able to reduce these emissions. The nature of $CH_4$ emissions through 'leaks' is episodic and spatially variable, making their monitoring a complex task, being partly addressed by atmospheric surveys with various types of instruments. Continuous records are preferable to snapshot surveys for monitoring a site, and one solution would be to deploy a permanent network of sensors. Deploying

such a network with research-level instruments being expensive, low-cost and low-power sensors could be a good alternative. However, low cost entails usually lower accuracy and the existence of sensors drifts and cross-sensitivity to other gases and environmental parameters. Here we present four tests conducted with two types of Figaro TGS sensors on a laboratory experiment. The sensors were exposed to ambient air and peaks of $CH_4$ concentrations. We assembled four chambers, each containing one TGS sensor of each type. The first test consisted in comparing parametric and non-parametric models to reconstruct the

$CH_4$ peaks signal from observations of the voltage variations of TGS sensors. The obtained relative accuracy is higher than 10% to reconstruct the maximum amplitude of peaks (RMSE $\leq$ 2 ppm). Polynomial regression and multilayer perceptron (MLP) models gave the highest performances for one type of sensor (TGS 2611C, RMSE = 0.9 ppm) and for the combination of two sensors (TGS 2611C + TGS 2611E, RMSE = 0.8 ppm) with a training set size of 70% of the total observations. In the second test, we compared the performance of the same models with a reduced training set. To reduce the size of the training

set, we have employed a stratification of the data into clusters of peaks that allowed us to keep the same model performances with only 25% of the data to train the models. The third test consisted of detecting the effects of age in the sensors after six months of continuous measurements. We observed performance degradation through our models, of between 0.6 and 0.8 ppm. In the final test, we assessed the capability of a model to be transferred between chambers on the same type of sensor, and found that it is possible to transfer models only if the target range of variation of $CH_4$ is similar to the one on which the model

was trained.





# 1 Introduction

Methane ($CH_4$) is a greenhouse gas 28 times more potent than carbon dioxide considering its warming potential over 100 years (Travis et al., 2020). Anthropogenic $CH_4$ emissions account for 60% of global emissions (Saunois et al., 2016). Fugitive leaks of natural gas at industrial facilities also present a safety hazard. Emissions from such facilities need to be continuously monitored, due to the episodic and spatially variable nature of leaks (Coburn et al., 2018). Leaks can be detected and quantified by LDAR surveys (Leak Detection And Repair) to detect high concentrations caused by a leak. Those surveys are periodical and have limitations related to the portability of instruments or accessibility of sites. A possible solution to overcome these limitations is to deploy a network of sensors that continuously measure methane concentrations around an emitting area (Kumar et al., 2015). Deploying such a network with highly precise instruments, using techniques such as cavity ring down spectrometry (CRDS) is, however, cost prohibitive. Low cost sensors such as low power metal oxide semiconductor (MOS) sensors for methane are an alternative. Recent studies (Riddick et al., 2020; Casey et al., 2019; Collier-Oxandale et al., 2018; Jørgensen et al., 2020; Rivera Martinez et al., 2021; Eugster et al., 2020) tested the ability of MOS sensors to monitor methane concentrations in natural and controlled conditions and showed a fair agreement between the concentrations derived from the sensors and those from high precision reference instruments. MOS sensors are composed of a semiconducting metal oxide sensing element heated at a temperature between 200° to 400 °C (Özgür Örnek and Karlik, 2012; Barsan et al., 2007). When the semiconducting material is in contact with an electron donor gas like $CH_4$, a change in the conductivity occurs, measured by an external electrical circuit (Özgür Örnek and Karlik, 2012). MOS sensors are known to be less precise than CRDS to $CH_4$ variations, although they can detect small variations in concentrations. Most MOS sensors have cross-sensitivities to other electron donors and to environmental variables such as absolute humidity, pressure and temperature (Popoola et al., 2018) with non-linear interactions (Rivera Martinez et al., 2021).

Biases affect $CH_4$ measurements derived from low-cost sensors because of cross sensitivities to other gases, dependence to environmental factors and internal drifts e.g. due to aging. Figaro Taguchi Gas Sensors (TGS) are a particular series of MOS capable of measuring $CH_4$. In order to limit biases of these sensors, several studies proposed a calibration model against a high precision reference instrument. Casey et al. (2019) compared different calibration approaches with inverse and direct linear models and artificial neural networks to quantify $O_3$ from an SGX Corporation MiCS-2611 sensor, CO from a Baseline Mocon PID sensor, $CO_2$ from an ELT S-100 non-dispersive infrared (NDIR) sensor and $CH_4$ from observations of a Figaro TGS 2600 sensors. Collier-Oxandale et al. (2018, 2019) applied multilinear models, including interactions from environmental variables, to predict $CH_4$ concentrations and to detect and quantify VOCs from Figaro TGS 2600 and TGS 2602 MOS sensors at two sites with active oil and gas operations. Eugster et al. (2020) used empirical functions and Artificial Neural Networks (ANN) to derive $CH_4$ concentrations from 6 years of data collected with Figaro TGS 2600 sensors at a field site in the Arctic. Riddick et al. (2020) derived nonlinear empirical relationships for Figaro TGS 2600 sensors from three experiments with durations varying from one day to one month. Rivera Martinez et al. (2021) reconstructed $CH_4$ concentrations variations in room air from Figaro TGS 2611-C00 sensors using ANN models and co-variations of temperature, water mole fraction and pressure.


Nevertheless, those comparisons were limited by the choice of a specific reconstruction model and restricted to only one type
of sensor.

There is a need for a more thorough comparison of different calibration approaches for Figaro MOS sensors applied to
measure $CH_4$. In addition, there is a need to assess the performances of MOS sensors to detect and quantify $CH_4$ spikes typical
of industrial emission. This study aims to compare several parametric (linear and polynomial) and non-parametric models
(random forest, hybrid random forest and ANN) applied to different combinations of Figaro TGS sensors to reconstruct the
$CH_4$ signals of repeated atmospheric spikes, based on the observed voltage of each sensor and other variables. The $CH_4$ signal
we aim to reconstruct is representative of variations observed in the atmosphere from leaks that occur within or close to an
emitting industrial facility, i.e. short duration $CH_4$ enhancements (spikes) lasting between 1 to 7 minutes and ranging between
1 to several tenth of ppm above an atmospheric background concentration of around 2 ppm (Kumar et al., 2021). In this study,
we performed a laboratory experiment where a CRDS instrument and many TGS sensors of different types were exposed to
a controlled air flow with artificially created $CH_4$ concentration spikes (section 2). The spikes were composed of pure $CH_4$
and did not contain any VOCs, although those species could be present in natural gas leaks from oil and gas facilities. The
experiment lasted four months and provided 838 spikes, which give us a dense and complex dataset to train and test different
models for reconstructing $CH_4$ variations. For low-cost sensors, a collocation is often required with a highly precise reference
instrument to train an empirical calibration model. This training phase should be as effective (parsimonious) as possible. The
strategy is to reduce the time and maintenance costs of having a reference instrument on site if the purpose is to bring it in
the field for future studies where low-cost sensors would have to be calibrated. We investigate the problem of 'parsimonious
training' by testing different configurations (model and inputs) to establish the minimum amount of reference data needed to
obtain good performances with the low-cost sensors (section 3.2 and 3.3). Secondly, since the performance of low-cost sensors
may change with time, it is important to understand if their measurements could be affected by a drift of their sensitivity
over time. We address this problem of 'non-stationary training' by comparing different calibration models for a second spike
experiment conducted six months after the first one (section 3.4). Thirdly, sensitivities may vary from one sensor to another
and may require a sensor-specific calibration model, which becomes a problem when a large number of sensors are deployed.
Finding a robust calibration model that could be trained using data from one or several sensors and applied to others remains
an open question. We bring some insight to this problem of 'generalized calibration' by training models to reconstruct the
$CH_4$ signal from a group of sensors located in the same chamber and applying them to other groups of sensors in a different
chamber (section 3.5). To assess the performance of the calibration models and particularly their capability to reconstruct
spikes of several ppm occurring upon a background $CH_4$ level, we define here as an acceptable performance to be an error
less than the 10% of the maximum amplitude of the peaks we aim to reconstruct. In our case, this requirement is an RMSE
of 2 ppm between the reconstructed $CH_4$ data from low cost sensors and the true data from a reference instrument at a time
resolution of 5 seconds.



## 2  Methods

### 2.1  Experimental set-up

#### 2.1.1  Low-cost $CH_4$ sensors

For the experiment, four independent sampling chambers were assembled. Each chamber contained a Figaro TGS 2600, TGS
2611-C00 and TGS 2611-E00 sensor, alongside a relative humidity and temperature sensor (DHT22 or Sensirion SHT75), and
a temperature and pressure sensor (Bosch BMP280, see Table 1 for details). Issues with the logger system produced gaps in
environmental variables data, thus observations information from an external chamber (E, see figure 1b and Table 1 for details)
was used in the correction of the sensitivity across all chambers. The sensors were disposed on a circuit board to minimise the
direct heating influence of the TGS sensors on temperature measurements. The sampling chamber was made of acrylic/glass
with a gas inlet and outlet and a port for the electrical cables (Figure 1a). Each sensor was connected in series with a high-
precision load resistor which controlled sensitivity (Figaro, 2013, 2005). The voltage across each load resistor was recorded
by an AB Electronics PiPlus ADC board, mounted on a Raspberry Pi 3b+ logging computer, sampling at a frequency of 0.5
Hz (2s). This voltage measurement was used in our characterisation algorithms, referred to hereafter as the sensor voltage. We
focus on the reconstruction of $CH_4$ using only the TGS 2611-C00 and TGS 2611-E00 data.

#### 2.1.2  Generation of methane spikes on top of ambient air

The experiment lasted 130 days from 28 October 2019 to 5 March 2020. During this period, the six chambers containing TGS
sensors sampled ambient air pumped from the roof of the laboratory. Relative humidity, air pressure and temperature were
measured in the ambient air flux, as well as $CH_4$ , using a Picarro CRDS G2401 reference instrument. No calibration was
considered on the CRDS instrument during the experimental period due to its high precision and low drift over time (less than
1 ppb per month; (Yver-Kwok et al., 2015)).

To expose the TGS sensor chambers to $CH_4$ enhancements (spikes) of different durations and amplitude comparable with
typical enhancements observed around industrial sites (Kumar et al., 2021), we designed an automatic system to add small
amounts of $CH_4$ on top of the ambient air acquired from our roof. The system presented on figure 1b consists an ambient
air flow to which was periodically added a small amount of a gas from a cylinder containing 5% of $CH_4$ (in argon), controlled
by two mass flow controllers denoted MFC1 and MFC2 in Figure 1b.

The occurrences of the spikes were programmed to be automatically generated, with at least three spikes each day. The
duration and magnitude of the spikes were predefined and controlled by varying the flows of MFC1 and MFC2, the two mass
flow controllers being programmed to add an amount of $CH_4$ to produce spikes of an expected amplitude ranging between 3
and 24 ppm. Two different types of spikes were generated. The first type, with large amplitudes between 20 and 24 ppm, was
generated from 28 October 2019 to 9 December 2019. The second type, with smaller amplitudes ranging between 5 ppm and
10 ppm but with a higher number of spikes during a given period of time, was generated from 9 December 2019 to 5 March
2020. The typical duration of the spikes of both types varied between 1 and 7 minutes, which is longer than the known response





**Table 1.** Summary of the sensors included on each logger box.

| Chamber | Figaro TGS | Temperature & Relative Humidity sensor | Temperature & Pressure sensor | Load Resistor |
|---|---|---|---|---|
| A | TGS 2600 TGS 2611-C00 TGS 2611-E00 | DHT22 | BMP280 | 50 KΩ |
| C | TGS 2600 TGS 2611-C00 TGS 2611-E00 | SHT75 | BMP280 | 50 KΩ |
| F | TGS 2600 TGS 2611-C00 TGS 2611-E00 | SHT75 | BMP280 | 50 KΩ |
| G | TGS 2600 TGS 2611-C00 TGS 2611-E00 | SHT85 | BMP280 | 50 KΩ |
| E | TGS 2600* TGS 2611-C00* TGS 2611-E00* | SHT75 & DHT22 | BMP180 | 5 KΩ |

* Two sensors of this type

time of the TGS sensors. Gas from the 5% $CH_4$ cylinder that persisted on the air flow after a spike in segment A-B (Fig. 1b) was expulsed though MFC2, preventing very high $CH_4$ concentrations to remain in the air flow following a spike. We verified

that the amount of gas with 5% of $CH_4$ added to the air flow measured by the TGS sensors did not affect the air pressure, temperature and relative humidity in the chambers.

The volume of each chamber is 100 ml and the flow rate through the chambers was fixed to 2.5 L per minute. We did not test the effect of increasing the flow rate on the TGS measurements. Instead, we decided to choose a high enough flow rate to reduce the buffering effect of the chamber volume that would systematically smooth the $CH_4$ spikes. Despite this set-up, a

buffering effect was still present in the chamber, evidenced by the fact that after stopping the injection of air with 5% $CH_4$ , the $CH_4$ draw down in the chamber was observed to be smooth and lagged the drawdown of the CRDS instrument by a time constant of 10s, consistent with previous measurements on buffer volumes acting as a low-pass filter (Cescatti et al., 2016).

To determine the time constant ($\tau$) of the buffer effect of the chambers, we applied an exponential weighted moving average to the CRDS data with different values of $\tau$ and compared them with the shape of the response of the TGS sensor (see Fig. A1).

A similar approach was employed by (Jørgensen et al., 2020) to compensate for effects of micro turbulent mixing of subglacial air with atmospheric observations. Before applying this temporal smoothing on the CRDS data, we resampled both signals, the reference CRDS and the TGS, from their original time resolutions (1s and 2s, respectively) to a common time resolution of 5s.





## 2.2 Separating CH$_4$ spikes from background variations in ambient air

Different algorithms have been proposed to identify short term variations of atmospheric signals (Ruckstuhl et al., 2012) from
slower variations of background variations in atmospheric composition. These approaches were applied to low-cost sensors
for the detection of local events (Heimann et al., 2015), and for the removal of diurnal periodical signals to identify peaks of
air pollutants (Collier-Oxandale et al., 2020). In this study, we want to separate the background of slowly varying CH$_4$ in the
outside air pumped from the laboratory roof from the signal of the CH$_4$ spikes by using an algorithm.

We followed a three-step approach. The first step was to remove the impact of H$_2$O variations on the sensor voltage signals,
given that H$_2$O changes in the background air. Previous studies (Eugster and Kling, 2012; Rivera Martinez et al., 2021)
demonstrated a direct dependence between the voltage/resistance of metal oxide sensors and H$_2$O concentration. In order to
determine this relationship, we used the background H$_2$O mole fraction and TGS voltage measurements in ambient air during
a period of 32 days with no CH$_4$ spikes, and regressed both variables to derive the H$_2$O-sensitivity of the voltage of each TGS
sensor in mV per ppm H$_2$O. This linear model of the Voltage - H$_2$O sensitivity was applied to voltage time series of the TGS
sensors during the spike measurement period.

The second step was to separate background and spike conditions from voltage variations in the time series. We tested two
approaches. The first approach applied the peak detection algorithm of (Coombes et al., 2003) to detect the voltage associated
to spikes and separate the background signal by a linear interpolation between non-spikes values at the start and the end of each
spike. The second approach applied the Robust Extraction of Baseline Signal (REBS) algorithm from Ruckstuhl et al. (2012)
to separate voltage observations associated to background from those during the spikes. The principle of REBS is to compute
local regressions over the time series on small moving time windows (60 minutes) and to iteratively identify outliers that are
far from the modelled background, based on a threshold. Here, the detected outliers are considered to belong to a spike. The
threshold or scale parameter, $\beta$, defines a range in number of standard deviations around the modelled baseline. A value of $\beta$
= 3.5 ppm was used. The third step was to remove observations corresponding to baseline and keep only the data classified as
spikes, which form the signal of interest in this study.

## 2.3 Modelling CH$_4$ spikes from TGS sensor voltages and environmental variables

The impact of different magnitudes of the variables used as predctors are prone to affect the parameters of the models in the
training stage. Thus, to reduce this impact we standardize the inputs before training the models. We chose a robust scaler
unaffected by outliers by removing the median and scaling the data to a quantile range (Demuth et al., 2014). To reconstruct
CH$_4$ spikes from TGS sensor voltages, we applied linear and polynomial regressions, ANN and Random Forest models, all
trained using the CRDS measurements. We assessed the performance of the different models using a k-fold cross validation,
here with k = 20. A fraction of the data was used for the training of each model and the rest for evaluation. We repeated this
training and evaluation process with a moving window to make a robust assessment of each model performance considering all
data available. We specified two cases for the relative sizes of the training and evaluation (test) sets. The first case used training
and test set fractions of 70% and 30% of the observations, respectively, and the second one used 50% and 50%. We focus in the



example below on the spike data from one chamber (chamber A) using different models as test inputs: 1) voltages of TGS C or E sensors separately, 2) voltages of a single sensor type and measurements of $H_2O$, temperature and pressure, 3) combined TGS C and E voltages, and 4) combined TGS C and E voltages, plus $H_2O$, temperature and pressure.

### 2.3.1 Linear and multilinear regression models

Linear regressions between dry air $CH_4$ concentrations from the CRDS and TGS sensor voltages are the simplest models, used in studies with similar low-cost sensors by others (Collier-Oxandale et al., 2018; Casey et al., 2019; Cordero et al., 2018; Spinelle et al., 2015, 2017; Malings et al., 2019). We derived linear regressions between the reference $CH_4$ from the CRDS instrument and the sensor voltage, as well as a multi-linear regression including voltage, $H_2O$, air pressure and temperature, as given by:

$$\hat{y}_{CH_4}(x_1 = V_{TGS}) = \beta_0 x_1 + \beta_1 \tag{1}$$

and

$$\hat{y}_{CH_4}(x_1 = V_{TGS}, x_2 = H_2O, x_3 = P_{Air}, x_4 = T_{Air}) = \alpha_1 x_1 + \alpha_2 x_2 + \alpha_3 x_3 + \alpha_4 x_4 + \alpha_5 \tag{2}$$

Where $\hat{y}_{CH_4}$ is the predicted methane concentration in ppm, $V_{TGS}$ the observed sensor voltage in V, $H_2O$ is the water vapor mole fraction in %, $P_{Air}$ the air pressure in kPa and $T_{Air}$ is the air temperature in °C.

### 2.3.2 Polynomial regression models

The second type of models are second degree polynomials, for which we considered as predictors either TGS sensor voltage alone, or TGS voltage plus environmental variables, as given by:

$$\hat{y}_{CH_4}(x_1 = V_{TGS}) = \beta_0 + \beta_1 x_1 + \beta_2 x_1^2 \tag{3}$$

and

$$\hat{y}_{CH_4}(x_1 = V_{TGS}, x_2 = H_2O, x_3 = P_{Air}, x_4 = T_{Air}) = \beta_0 + \beta_1 x_1 + \beta_2 x_2 + \beta_3 x_3 + \beta_4 x_4 + \beta_5 x_1^2 + \beta_6 x_1 x_2 + \beta_7 x_1 x_3 +$$

$$\beta_8 x_1 x_4 + \beta_9 x_2^2 + \beta_{10} x_2 x_3 + \beta_{11} x_2 x_4 + \beta_1 2 x_3^2 + \beta_{13} x_3 x_4 + \beta_{14} x_4^2 \tag{4}$$

### 2.3.3 Random forest and hybrid random forest models

Random forest regressors (Breiman, 2001) are an ensemble learning method consisting of creating several decision trees to fit complex data. Each tree is composed of leaves defined hierarchically based on thresholds that group values of input variables,





constructed from a subset of predictors randomly chosen, a process known as 'feature bagging'. The prediction is made by
averaging the outputs of all the trees. As a non-parametric method, the generalization of Random Forests is limited by the range
of values present in the training set. A methodology proposed by Malings et al. (2019) to boost the generalization of random
forest models is to 'hybridize' them with a parametric model to be able to predict values that are out of those present in the
training set. The principle of hybridization consists in training a random forest model with about 80 to 90% of the observations,
and reserving 20 to 10% of the higher observations to train a linear or polynomial model. This approach allows us to benefit, on
one hand, from the capability to derive nonlinear relationships from the inputs, while on the other hand boosting the prediction
outside of the range present in the training set, here with a linear or polynomial model. Here we used both traditional and hybrid
random forest models. For the hybrid models, we reserved, for each cross-validation fold, the higher 10% concentrations to
train a polynomial fit, the remaining observations being fitted by the random forest. The same four cases of input combinations
explained in section 2.3 were used for the training of traditional and hybrid random forest models.

### 2.3.4 Artificial neural networks (ANN)

In recent studies with low-cost sensors (Rivera Martinez et al., 2021; Casey et al., 2019) ANN models have proven to be
powerful models to derive $CH_4$ concentrations from sensor signals. We chose here a Multilayer Perceptron (MLP) model due
to its ability to provide a universal approximator (Hornik et al., 1989) and generalization capabilities (Haykin, 1998). No prior
knowledge of relationships between variables is required to produce model outputs. Our MLP is composed of a series of units
(neurons) arranged in fully connected layers, each unit being a weighted sum of its inputs to which an activation function (tanh,
ReLU) is applied. The last layer of the network when used as a regressor usually has one unit and a linear activation function.
As a supervised learning algorithm, MLP requires examples (the training set) and an iterative learning algorithm to adjust the
weights of its connections. The main challenges for training an MLP are: 1) underfitting, when the model is not able to fit the
training set and 2) overfitting, when the model is not capable of generalizing new examples. Underfitting can be mitigated by
increasing the complexity of the MLP, and overfitting can be partly mitigated by weight decay regularization or early stopping
(Bishop, 1995; Goodfellow et al., 2016).

We built different MLP models using the BFGS algorithm (Bishop, 1995). The optimal number of layers and units was
determined using a grid search technique (Géron, 2019), resulting in 1) 4 hidden layers when using the voltage from a single
TGS sensor as input, with 2, 3, 5 and 2 units per layer, 2) when using TGS sensor voltages and other variables, 4, 2, 5 and 5
units per layer, 3) 5 layers when combining both TGS sensor voltages together (5, 3, 5, 5 and 4 units), and 4) 4 layers when
using both TGS sensor voltage types and other variables (5, 3, 5 and 5 units). The ReLU activation function was used on units
of the hidden layer and early stopping was used to prevent overfitting.

### 2.4 Finding a parsimonious model training strategy

To determine the minimum number of training observations to obtain a model with satisfactory performances, given our 2 ppm
RMSE requirement posed in the introduction, we followed a two-step approach. First, we stratified the data into different types
of spikes using an unsupervised hierarchical clustering algorithm (Johnson, 1967). Secondly, we constructed training sets by





**Table 2.** Percentage of spikes in each cluster (C1 to C9) considered for training different models.

|  | C1 | C2 | C3 | C4 | C5 | C6 | C7 | C8 | C9 | # Spikes in total | % of data in the training set |
|---|---|---|---|---|---|---|---|---|---|---|---|
| **Case 1** | 70% | 70% | 70% | 70% | 70% | 70% | 70% | 70% | 70% | 587 | 70.0% |
| **Case 2** | 70% | 10% | 10% | 10% | 10% | 10% | 10% | 10% | 10% | 122 | 12.5% |
| **Case 3** | 10% | 70% | 10% | 10% | 10% | 10% | 10% | 10% | 10% | 150 | 13.7% |
| **Case 4** | 10% | 10% | 70% | 10% | 10% | 10% | 10% | 10% | 10% | 147 | 14.5% |
| **Case 5** | 10% | 10% | 10% | 70% | 10% | 10% | 10% | 10% | 10% | 198 | 19.3% |
| **Case 6** | 10% | 10% | 10% | 10% | 70% | 10% | 10% | 10% | 10% | 105 | 17.8% |
| **Case 7** | 10% | 10% | 10% | 10% | 10% | 70% | 10% | 10% | 10% | 166 | 18.5% |
| **Case 8** | 10% | 10% | 10% | 10% | 10% | 10% | 70% | 10% | 10% | 150 | 16.7% |
| **Case 9** | 10% | 10% | 10% | 10% | 10% | 10% | 10% | 70% | 10% | 94 | 11.9% |
| **Case 10** | 10% | 10% | 10% | 10% | 10% | 10% | 10% | 10% | 70% | 124 | 24.5% |
| **Case 11** | 10% | 10% | 10% | 10% | 10% | 10% | 10% | 10% | 10% | 84 | 10.0% |

randomly selecting spikes inside each cluster, in different proportions given in Table 2, and evaluated our models against the remaining spikes used as a test set. This evaluation strategy helped us to understand the clusters that have the most influential impact on to increase the models performance. This allowed us to reduce the length of the training set by sampling the training
data preferentially in the most influential clusters.

The clusters of spikes were defined using the ward distance to determine a matrix measuring the degree of similarity between spikes using Dynamic Time Warping (DTW) (Sakoe and Chiba, 1978), and to construct a dendrogram. A threshold on the dendrogram allowed us to determine 9 different clusters from our dataset. For the second step, we defined 11 cases to construct training sets. Cases 1 and 11 correspond to sampling 70% and 10% of the data for training, respectively, equally distributed
across the clusters. Cases 2 to 10 correspond to preferential sampling one cluster over the others for training, by selecting 70% of the spikes in this cluster and 10% in all others. The purpose of this stratified data selection is to determine the type of spikes that best allows for reconstruction of the variations of $CH_4$ when training a model. At this stage we are not interested in the temporal dependency between observations since we train models with instant values. On a practical application side, a parsimonious model training strategy will require users to expose their sensors to specific type of 'highly influential' spikes on
a shorter period from, e.g., a laboratory experiment like the one described above, then train the models upon those spikes and apply them to data collected in the field.

## 2.5 Assessing ageing effects of the sensors

To assess the effect of ageing sensors on the reconstruction of $CH_4$, we conducted a 33-day experiment from 11 August to 12 September 2020, six months after the first experiment described in section 2.1.2. The spike generation system was the
same. Between the two experiments, the chambers containing TGS sensors had been measuring ambient air pumped from our





laboratory roof. To assess the ageing effect on the TGS sensors during the six-month interval, we selected the two models that gave the highest performance for the first experiment and applied them to simulate the spikes generated during the second experiment.

## 2.6 Finding generalized models that can be used for other sensors of the same type

We were interested in understanding to what extent a model trained with the outputs of a given TGS sensor type in a given chamber could be applied to other sensors of the same type in other chambers. The experiment consisted of training a model per sensor and chamber with the best configuration subset based on the cluster classification outlined in section 2.4. The trained model is then used to reconstruct the $CH_4$ spikes using data from the TGS in other chambers and compare their performances. For this, we used data from chambers A, C, F and G to train chamber-specific models, and used each chamber-specific model 250 to reconstruct $CH_4$ spikes using data from other chambers, as shown in Table 1. The four chambers have a load resistor of 50 k$\Omega$ and contain three TGS sensors each. We did not use data from chamber D and E because it has a load resistor of 5 k$\Omega$ and on the chamber E contains two of each TGS sensor.

## 2.7 Metrics for performance evaluation

The performance of the models to reconstruct the dry $CH_4$ concentrations observed by the CRDS instrument using TGS sensors 255 was assessed using a decomposition of the mean squared deviation (MSD) of the misfits between reconstructed and true $CH_4$ (Kobayashi and Salam, 2000), to separate the main source of errors when comparing different models. MSD was decomposed into the sum of the Square Bias (SB), the difference in the magnitude fluctuation (SDSD) and the lack of positive correlation weighted by the standard deviation (LCS). A large SDSD indicates an incorrect reconstruction of $CH_4$ spike magnitudes. A large LCS indicates an incorrect reconstruction of spike phase or shape. The equations for each error term according to 260 Kobayashi and Salam (2000) are given by:

$$\text{SB} = (\overline{\hat{y}_{CH_4}} - \overline{y_{CH_4}})^2 \tag{5}$$

$$\text{SDSD} = (\sigma_{\text{Model}} - \sigma_{\text{Ref}})^2 \tag{6}$$

$$\text{LCS} = 2\sigma_{\text{Model}}\sigma_{\text{Ref}}(1 - \rho) \tag{7}$$

$$\text{MSD} = \text{SB} + \text{SDSD} + \text{LCS} \tag{8}$$

With $\overline{\hat{y}_{CH_4}}$ the mean of the prediction, $\overline{y_{CH_4}}$ the mean of the reference observations, $\sigma_{\text{Model}}$ the standard deviation of the modelled $CH_4$ time series, $\sigma_{\text{Ref}}$ the standard deviation of the reference one and $\rho$ their correlation coefficient. All results presented below are using metrics computed for the test set only.





## 3 Results

### 3.1 Data pre-processing and baseline correction

Figure 2 shows the pre-processing steps of the dataset, with the identification and removal of the background signal from the spikes in the time series. We removed outliers and the first 30 minutes of observations in case of a reboot of the data loggers of each chamber, i.e. during stabilization of the sensors. The original observations on a time step of 2 s were resampled to means of 5 s. Time shift due to incorrect clock synchronization between the reference CRDS instrument and the loggers of the TGS sensors were partly corrected with a search of the maximum correlation on non-overlapping windows of 6 hours and a manual

inspection of the agreement between TGS voltages and $CH_4$ observations of the reference CRDS instrument.

Environmental variables ($H_2O$, temperature, pressure) were filtered using a low pass filter (Press and Teukolsky, 1990) to remove high frequency noise from the sensors and circuit connections. The water vapour mole fraction was calculated with Rankine's formula (Eq. 9) from relative humidity (RH) in % and temperature (T) in °C from the DHT22 sensors and pressure (P) in Pa from the BMP180 sensors in each chamber, according to:

$$H_2O_{Mole\ Fraction} = 100 \times \left( \frac{\frac{RH}{100} \times e^{\frac{13.7 - 5120}{T + 273.15}}}{\frac{P}{100000} - \frac{RH}{100} \times e^{\frac{13.7 - 5120}{T + 273.15}}} \right) \tag{9}$$

An example of several spikes obtained after the pre-processing and background signal removal is shown in figure 3 (see also figure A3). The entire spike dataset contains 838 spikes, representing 1.6% (35536 5s observations) of the full dataset.

### 3.2 Reconstruction of $CH_4$ spikes

Figure 4 shows the reconstruction of several spikes by the linear, polynomial, random forest (RF), random forest hybrid (RFH)

and MLP models using data from the type C TGS sensor in chamber A. Figure 5 shows the reconstruction results using data from the type E TGS sensor in chamber A. In both figures, the model training set contains 70% of the total observations available. The spikes reconstructed by the different models show good agreement with the reference $CH_4$ signal for the type C sensors, but not for type E ones which are associated with phase errors and greater noise in the reconstructed $CH_4$. The linear model, RF and the RFH models broadly capture the mean amplitude of spikes, but they are less capable of reconstructing small

$CH_4$ variations on the top of the spikes. The RF and RFH models (the latter with a polynomial model) provided very similar outputs, with a small enhancement of the amplitude for RFH during some spikes and noise, especially with type E sensors (Fig. 5). The MLP model showed a constant underestimation of the spike magnitudes and produced smoother spike shapes, presenting a low pass filter behaviour. The polynomial fit models appeared to perform better. Despite the phase misfit of models with type E sensors, for all the models, both type C and type E meets our requirement target of an RMSE $\leq 2$ ppm (MSD $\leq 4$

ppm$^2$). With a stricter requirement of an error less than the 5% of the maximum amplitude of the peaks (RMSE $\leq 1$ ppm) only Type C is adequate.

Figure 6 shows the distributions of the correlations ($\rho$) between modelled and observed $CH_4$ spikes for the 20-fold validations periods (test sets) for different models. We distinguished two groups of models, based on median values of $\rho$. The first group





corresponds to models trained with type E sensors data only, characterized by $\rho_{\mathrm{Median}} \leq 0.93$. The second group corresponds to

models trained with type C sensor data only, or with data from both types of sensors, characterized by a higher $\rho_{\mathrm{Median}} \geq 0.96$. Among the models in the first group, the Polynomial Model gave the largest correlations ($\rho_{\mathrm{Median}} = 0.92$, interquartile range (IQ) = 0.001). Among the models in the second group, the Polynomial Model also showed the largest correlation, especially with both types of sensors, and a training set of 50% of the observations ($\rho_{\mathrm{Median}} = 0.98$, IQ = 0.004), closely followed by the MLP model with the same inputs and the same training set size ($\rho_{\mathrm{Median}} = 0.98$, IQ range=0.006). The Random Forest,

Random Forest Hybrid and MLP models also showed high correlations when input data are from the type C sensors and the training uses 70% of the observations (RF $\rho_{\mathrm{Median}} = 0.982$, RFH $\rho_{\mathrm{Median}} = 0.983$ and MLP $\rho_{\mathrm{Median}} = 0.982$). These three models however had lower correlations when input data are from type E sensors (RF $\rho_{\mathrm{Median}} = 0.893$, RFH $\rho_{\mathrm{Median}} = 0.894$ and MLP $\rho_{\mathrm{Median}} = 0.92$). Phase errors were reduced when training the models with either type C sensor data or data from both sensors. The length of the training set had an important impact on the spread of the correlations across the 20-fold periods.

With 70% of observations in the training set, the IQ of the correlations increased, whereas for a smaller training set, the IQ was smaller but the distribution of the correlations showed more outliers. The inclusion of environmental variables (Fig. 6b) as input to models, in addition to voltages from TGS sensors, reduced significantly the phase error in the Random Forest models but produced little improvements in the results from other models.

    Figure 7 shows the MSD error decomposition for the different models and for the two training set sizes of 70% and 50%,

respectively. We observed that the LCS component of the MSD (related to a phase misfit of the modelled series) is the principal source of error across the different models, regardless of the input used or the size of the training set, meaning that models have more difficulties to reproduce the phase of the spikes than their amplitude. A systematically higher LCS error was obtained when data from type E sensors are used as input, and there is also a larger SDSD error with this type of sensor. For example, the largest LCS error was found with a training set of 70% for the Random Forest models ($\mathrm{LCS}_{\mathrm{RF}} = 4.67$ ppm$^2$, $\mathrm{LCS}_{\mathrm{RF}}$=1.24

ppm$^2$, $\mathrm{LCS}_{\mathrm{RF}}$=0.79 ppm$^2$ with type E sensor data, type C sensor data and both types respectively) as well as for the RFH models, when compared with other models. Additionally, the inclusion of environmental variables had little effect on the model performance. This was clearly shown for the LCS error of the polynomial model, for a training set of 70% of the data, which was identical with and without environmental variables as input ($\mathrm{LCS}_{\mathrm{poly}}$=3.0 ppm$^2$ for the type E sensor, $\mathrm{LCS}_{\mathrm{poly}}$=0.84 ppm$^2$ for the Type C sensor and $\mathrm{LCS}_{\mathrm{poly}}$=0.7 ppm$^2$ for both types). Reducing the size of the training set affected mostly the SDSD

component, by slightly lowering the capability of models to reconstruct the amplitude of the CH$_4$ spikes. For the non-parametric models, reducing the size of the training set also increased the bias error (SB), an effect that was partially mitigated with the inclusion of environmental variables. Amongst the non-parametric models, the MLP obtained similar performance than the parametric polynomial model ($\mathrm{MSD}_{\mathrm{MLP}}$= 3.2 ppm$^2$, $\mathrm{MSD}_{\mathrm{MLP}}$=0.85 ppm$^2$ and $\mathrm{MSD}_{\mathrm{MLP}}$=0.7 ppm$^2$ for type E sensors, type C sensor and both types together, respectively). To summarize, the choice of the sensor type used to train the models affected

more the reconstruction error than the selection of the model. The type C sensor data produced the lowest error compared to type E, irrespective of the model used. Overall, the polynomial model gave better performance than the non-parametric models. More detailed statistics are summarized in tables A1 and A2.



### 3.3 Results of parsimonious training tests

Figure 8 shows the result of the spike clustering. Based on spike similarity, we found 9 clusters. The peaks with short durations
(under 50s) and containing only one spike were grouped into cluster C1 (signal amplitude (sa) $\leq$ 6 ppm) and cluster C3 (6
ppm $\leq$ sa $\leq$ 12 ppm). Peaks with longer duration (over 50 s) were grouped in clusters C2 (sa $\leq$ 4 ppm) and C4 (4 ppm $\leq$ sa $\leq$
8ppm). Peaks with very long duration (between 50s to 1.5 min) were grouped in cluster C5. Peaks with a small concentration
at the beginning (around 6 ppm) followed by a larger peak (up to 12 ppm) were grouped in cluster C6. Peaks with larger
concentrations ($\geq$ 12 ppm) and complex in shape were grouped in clusters C7, C8 and C9, respectively. The cluster regrouping
the largest number of spikes (191) was C4, and the one with the smallest number of spikes (17) is C8.

Figure 9 shows the error of the models against the test set, for each of the training cases listed in Table 2, based on spikes
chosen from different clusters for doing the training (see Section 2.4). The results are summarized in Tables A3 and A4. First,
the polynomial and MLP models performed consistently better than the other models, the MLP being slightly better for most
of the cases. In contrast, the linear, random forest and random forest hybrid models had the highest error, regardless of the
sensor type or the addition of environmental variables. To compare the performances of the models trained by spikes from
different clusters (Table 2), we ranked them by their error. The MLP model with type C sensor data as input, and training with
spikes from Case 10 (124 spikes) produced the smallest error (MSD = 0.79 ppm$^2$), followed by the same model for Case 8
(MSD = 0.85 ppm$^2$, 150 spikes), Case 9 (MSD = 0.86 ppm$^2$, 94 spikes) and Case 11 (MSD = 0.87 ppm$^2$, 84 spikes). For the
MLP model, Case 4 (147 spikes), Case 1 (587 spikes) and Case 7 (166 spikes) performed slightly worse, with a MSD = 0.89
ppm$^2$. Finally, Case 2 (MSD = 0.9 ppm$^2$, 122 spikes), Case 3 (MSD = 0.91 ppm$^2$, 150 spikes), Case 6 (MSD = 0.93 ppm$^2$,
spikes) and Case 5 (MSD = 0.95 ppm$^2$, 198 spikes) showed lower performances. From the model ranking, we derived
the following conclusions. Firstly, the smallest error did not correspond to the most parsimonious training set (Case 11) but
to a larger training set (Case 1, 70% of the data). Nevertheless, we found that Case 11, which was constructed with an even
selection of spikes from all the clusters, each in a modest proportion (10% from each cluster) provided better performance than
most of the other training cases. This result shows that some clusters introduce less information or have redundancy. Overall,
the best performances corresponded to Cases 10, 8 and 9, which all included spikes with complex shapes from clusters C7, C8
and C9. Training models with a sample of those spikes thus ensured better model performances.

### 3.4 Results for possible ageing effect on model performance

To test for a possible ageing effect of the sensors, we selected the two best models (polynomial regression and MLP) found
in the previous section, and trained them following the best training configuration (Case 10). After being trained using data
from the first experiment, these two models were applied to reconstruct the spikes of the second experiment, six months later.
A summary of the results is presented on Table 3. We observed that after six months, the RMSE error produced by the models
increased from 0.57 to 0.85 ppm. The models trained with type E sensor data showed a smaller degradation (higher RMSE)
after six months compared to those trained with the type C sensor data. Considering the amplitude of the peaks that we aim
to reconstruct ($\sim$ 24 ppm), a possible drift caused by ageing effects on the sensors appeared to be a small source of error in





**Table 3.** Comparison of error for reconstructing spikes in experiment 2, using the two best models (polynomial and MLP) trained with the best training set configuration during experiment 1.

|  | Mean RMSE$_1$* (ppm) | Mean RMSE$_2$** (ppm) | Difference (ppm) | Monthly RMSE increase (ppm month$^{-1}$) |
|---|---|---|---|---|
| **Poly (C)** | 0.96 | 1.82 | 0.85 | 0.14 |
| **MLP (C)** | 0.95 | 1.75 | 0.80 | 0.13 |
| **Poly (E)** | 1.84 | 2.53 | 0.69 | 0.11 |
| **MLP (E)** | 1.84 | 2.41 | 0.57 | 0.09 |
| **Poly (C+E)** | 0.89 | 1.58 | 0.69 | 0.11 |
| **MLP (C+E)** | 0.86 | 1.51 | 0.64 | 0.10 |

\* For spikes reconstructed during experiment 1

\** For spikes of experiment 2 reconstructed with models trained on experiment 1

**Table 4.** Summary of spikes, observations and clusters detected following the procedure explained on Section 2.4 for chambers A, C, F, and G.

| Chamber | Number of Observations | Number of Spikes | Number of Clusters |
|---|---|---|---|
| **A** | 35536 | 836 | 9 |
| **C** | 35499 | 902 | 7 |
| **F** | 50089 | 861 | 9 |
| **G** | 50569 | 612 | 9 |

the reconstruction of CH$_4$ spikes during the second experiment. Assuming that the error of the sensors increased linearly with time, we determined an error 'drift rate' by computing the ratio of the difference in the error from both experiments divided by the time between them. We observed that for all the cases, the difference in the error is less than 1 ppm after six months and the mean RMSE on the second experiment is less than 2 ppm in all cases, except for the models trained with only the type E

sensor. Thus, even with aging the type E sensors would still meet our requirement of a RMSE smaller than 2 ppm. This shows the capability of our models to reconstruct spikes despite possible ageing effects of the sensors.

### 3.5 Generalized models

In this section, we address the comparison of model performances when we train a model on a subset of the sensor data from one chamber and reconstruct the spikes of the other chambers. Table 4 presents a summary of the number of spikes,

observations and clusters analysed for each chamber. The number of clusters, as well the number of spikes, were not equally captured by all the chambers. Only three chambers, A, F and G, shared the same number of clusters. Chambers C had a more limited number of peaks, due to a reduced sampling period.





To illustrate the performance of models for their ability to be generalized from one chamber to another, we selected the polynomial model with input data from the type C sensor (Fig. 10) and from the type E sensor (Fig. 11). The same results with the MLP model are shown in figures A7 and A8, respectively. The data in Figure 10 indicate that the error was lower for the test set of the chamber on which the model was trained than for the test sets of other chambers, as expected. In Figure 10a, c and d, we observed that the models trained with the data from chambers A, F or G produced good performances for reconstructing the spikes of another chamber, and met the requirement target of an RMSE $\leq$ 2 ppm. The models trained with the data from chamber C (fig 10b) however, performed poorly in reconstructing the spikes from the other chambers and met the target requirement only when trained using data from the same chamber. The performances of the MLP model were similar to those of the polynomial model in terms of generalization from one chamber to another. When trained by data from the type E sensor, our models were found to be less transferable from one chamber to another, meaning they had a larger error for the test sets of another chamber than for the one used for training (Figure 11). We inferred that the reconstruction of spikes from models of other chambers needs to be coherent with the number of clusters of the chamber used for training in order to ensure transferability of the models. This is the case for chambers A, F and G for which nine clusters were detected and the distribution of peaks within the clusters was similar (Figure 8, A9 and A10). On the other hand, if the clusters are not similar between chambers, the transferability of models is lower.

## 4 Discussion

Our results show that a pre-processing of the data to remove $H_2O$ effects and separate spikes from ambient air $CH_4$ variations, followed by a careful definition of the training set provides capabilities for different models to reconstruct the $CH_4$ spikes on a 5 s time step, across a large range of concentration variations and spike durations, meeting our requirement o a target error of RMSE $\leq$ 2 ppm. The TGS 2611-E00 (Type E) was the sensor with the poorest performance, regardless of the model employed, or of the subset of data used to train models, as shown by our tests with 5 chambers, each containing 5 different sensors. The model performances for TGS 2611-E00 were thus always poorer than for TGS 2611-C00 (Type C), with a degradation in the reconstruction coming from the larger misfit of the phase of the spikes signal than with the TGS 2611-E00 sensors. This probably is related to the carbon filter that is integrated within this type of sensor, to improve the selectivity. An additional step of the pre-processing algorithm could help to correct problems due to the carbon filter. The MSD error decomposition showed that the sources of error in the reconstruction were mainly from an inaccurate reconstruction of the phase, followed by a misfit of the magnitude of the spikes. The inclusion of environmental variables reduced the LCS component of the MSD, especially for non-parametric models. Nevertheless, for the Type E sensor, adding environmental variables increased the error in the reconstruction of the magnitude. Finally, we found that the error always increased with the reduction of the length of the training set, as previously shown by Rivera Martinez et al. (2021). This sensitivity to the training set mainly affected the non-parametric models due to their limited capability of extrapolation and their requirement of large datasets to keep good performances.



**How do our approach and results compare with previous studies?** Malings et al. (2019) performed a comparison of different calibration approaches including linear, quadratic, Gaussian models, clustering models, ANN and hybrid random forest models across low-cost sensors measuring different species ($CO_2$, CO, $NO_2$, $SO_2$ and NO) with the aim to calibrate Real-time Affordable Multi-Pollutant monitors (RAMP) to assess the air quality within a city, using a network of sensors. Their set of sensors included an NDIR $CO_2$ sensor, an Alphasense photoionization detector and an Alphasense electrochemical unit.

They found that a quadratic regression and a hybrid RF model produced the best performance across different pollutants for training sets with durations between 21 and 28 days, and observations with a resolution of 15 minutes. Our results showed that the hybrid random forest model did not perform as well as the polynomial model or the MLP for the reconstruction of $CH_4$ spikes using data from TGS sensors, and that these models were sensitive to the length of the training set for the k-fold cross validation. An improvement of our models' performances could be achieved with a selection of the proportion of observations

used for the parametric model. Nevertheless, the polynomial model gave consistently better results regardless of the inclusion of environmental variables.

    Casey et al. (2019); Rivera Martinez et al. (2021) and Eugster et al. (2020) used ANN models to derive $CH_4$ concentration from observations of TGS sensors and obtained good performances. Casey et al. (2019) suggested that the inclusion of correlated species (e.g. $CO_2$) rather than the type of sensor led to better performance for their MLP model to reconstruct $CH_4$.

The performance of their ANN model to reconstruct $CH_4$ variations provided an RMSE of 0.13 ppm for a range of variation between 1.5 and 4.5 ppm. Eugster et al. (2020) also found that the inclusion of other driving variables could increase the performance of ANN models. Their overall model performances for seven years of continuous $CH_4$ monitoring on ambient air in northern Alaska (range of variation between 1.7 and 2.1 ppm) with a Figaro TGS2600 gave an RMSE of the residuals of 0.043 $\mu$mol mol$^{-1}$ (0.69 ppm). Our results showed that different types of TGS sensors used with the same model gave

complementary information by reducing the error of the reconstruction and should be used, especially with non-parametric models. The performance of our best model for $CH_4$ spikes with concentrations much larger than those measured by Eugster et al. (2020), produced under controlled laboratory conditions, provides a mean RMSE of 0.9 ppm for a range of $CH_4$ variation between 3 and 24 ppm, thus rather comparable results. Regarding the calibration strategy, the clustering approach allowed us to determine nine clusters of spikes in our dataset, with three of them regrouping the largest peaks with complex shapes. This

classification allowed us to understand the impact of each cluster in the training. Cluster 9, composed with peaks of complex shape and a range of variation between 3 and 24 ppm was the one that provided the best information for training the models, due to the fact that spikes present on this cluster include information of larger and shorter peaks, medium peaks and larger peaks with patterns on top of the peaks. With the parsimonious training using Case 10, corresponding to a high proportion of peaks from cluster 9, we were able to reduce the length of the training dataset from 70% to 25% while maintaining similar

performance. This approach has a strong potential to reduce the length of the training set by selecting only observations from specific clusters defined from the data, and which represent the entire dataset.

    Concerning the ageing effect of the sensors, after six months, we observed only small increases in the RMSE of our models, between 0.6 to 0.8 ppm corresponding to an error increase rate of 0.1 ppm per month. Our results also showed the capability to





transfer the models from one chamber to another, provided that the chamber used for testing contains data with the same range

of $CH_4$ variations as the chamber used for training, which can be assessed by our clustering analysis of the data.

## 5   Conclusions

We performed a systematic comparison of different parametric and non-parametric models to reconstruct atmospheric $CH_4$ spikes under laboratory conditions, based on the voltages recorded by low cost metal oxide sensors. Other environmental variables such as temperature, pressure and water vapor were used. The true $CH_4$ time series comes from a high precision

instrument run alongside the low-cost sensors. The best models were a 2nd-degree polynomial function and a multi-layer-perceptron model. These two models both meet our requirements of a RMSE smaller than 2 ppm. We found that the main limitation was the large fraction of data (70%) needed to train the model. This would limit the use of low-cost sensors in the field, as they would need to be frequently trained with an expensive instrument at the same location. This limitation was partly overcome by adopting a stratified training strategy, namely to perform the training on fewer but more influential spikes selected

into orthogonal clusters applied to the whole dataset. This parsimonious training allows to use only 25% of the data to keep a model performance compliant with our 2 ppm RMSE threshold. We also showed that sensors' ageing effects after six months did not degrade too much the performances of our models. Finally TGS 2611C-00 was superior to TGS 2611-E00 model. For this experiment we generated about 800 peaks with some predefined shapes, future implementations should consider increase the diversity of shapes and durations of the generated peaks. Regarding the models employed, we assessed the performances

of models that considers no time dependency in the signal, more complex models that allows to include the time dependence such as Recursive neural networks (RNN) should be tested.

*Author contributions.* Olivier Laurent and Ford Cropley designed the Figaro® logger and conducted the laboratory experiments, Rodrigo Rivera and Diego Santaren developed the $CH_4$ reconstruction models. Rodrigo Rivera, Olivier Laurent and Cécile Mallet developed the baseline correction methodology of TGS sensors. Rodrigo Rivera, Gregoire Broquet and Philippe Ciais prepared the manuscript with collab-

oration of the other co-authors.

*Competing interests.* The authors declare that they have no conflict of interest.

*Acknowledgements.* This work was supported by the Chaire Industrielle Trace ANR-17-CHIN-0004-01 co-funded by the ANR French national research agency, Total Energies-Raffinage Chimie, SUEZ - Smart & Environmental Solutions and THALES ALENIA SPACE.



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



(a)  (b)

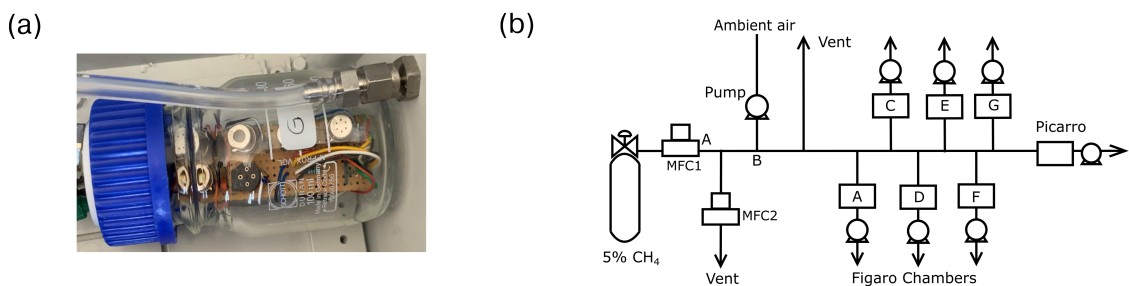

**Figure 1.** (a) Example of a chamber with three sensors inside, (b) Scheme of the spike creation experiment.





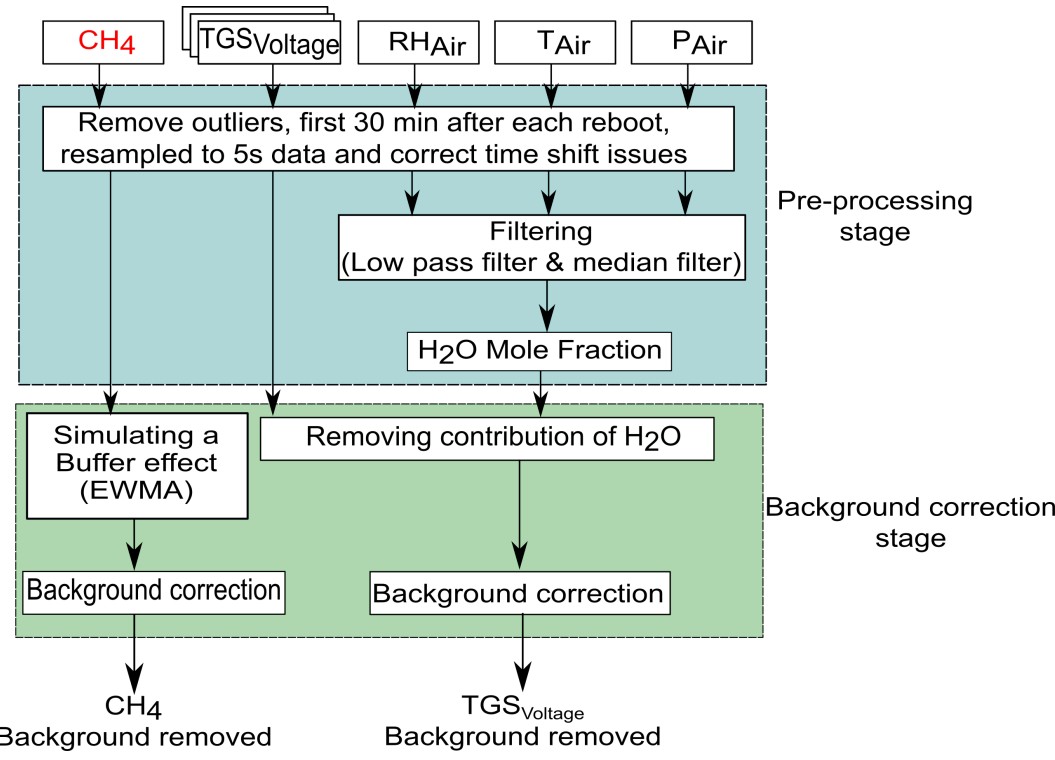

**Figure 2.** Data pre-processing diagram, correction of $H_2O$ effects, and separation of the spikes from background data in the time series.





**Figure 3.** (top to bottom) Time series of reference $CH_4$ signal from CRDS, voltage from TGS sensors, $H_2O$, temperature and pressure during a period of 30 minutes, after removing from the time series the variations of background signals, and applying the $H_2O$ correction to the voltage signal of TGS sensors. Dots on panels represent actual observations and lines between dots are drawn to show the shape of the signals.



**Figure 4.** Example of reconstruction of the CRDS reference $CH_4$ signal on a time step of 5 s for a few spikes in the test set by (a) a Linear model, (b) a polynomial model, (c) a Random Forest model, (d) a Random Forest Hybrid model and (e) a Multi-Layer Perceptron model trained with 70% of data and using as input data from the TGS 2611-C00 sensors only. The right panels show scatter plots between the reference $CH_4$ signal and the modelled outputs. The colour code is the density of observations.

TGS 2611-E00

**Figure 5.** Same as Figure 4 but for data from the TGS 2611-E00 sensors only.

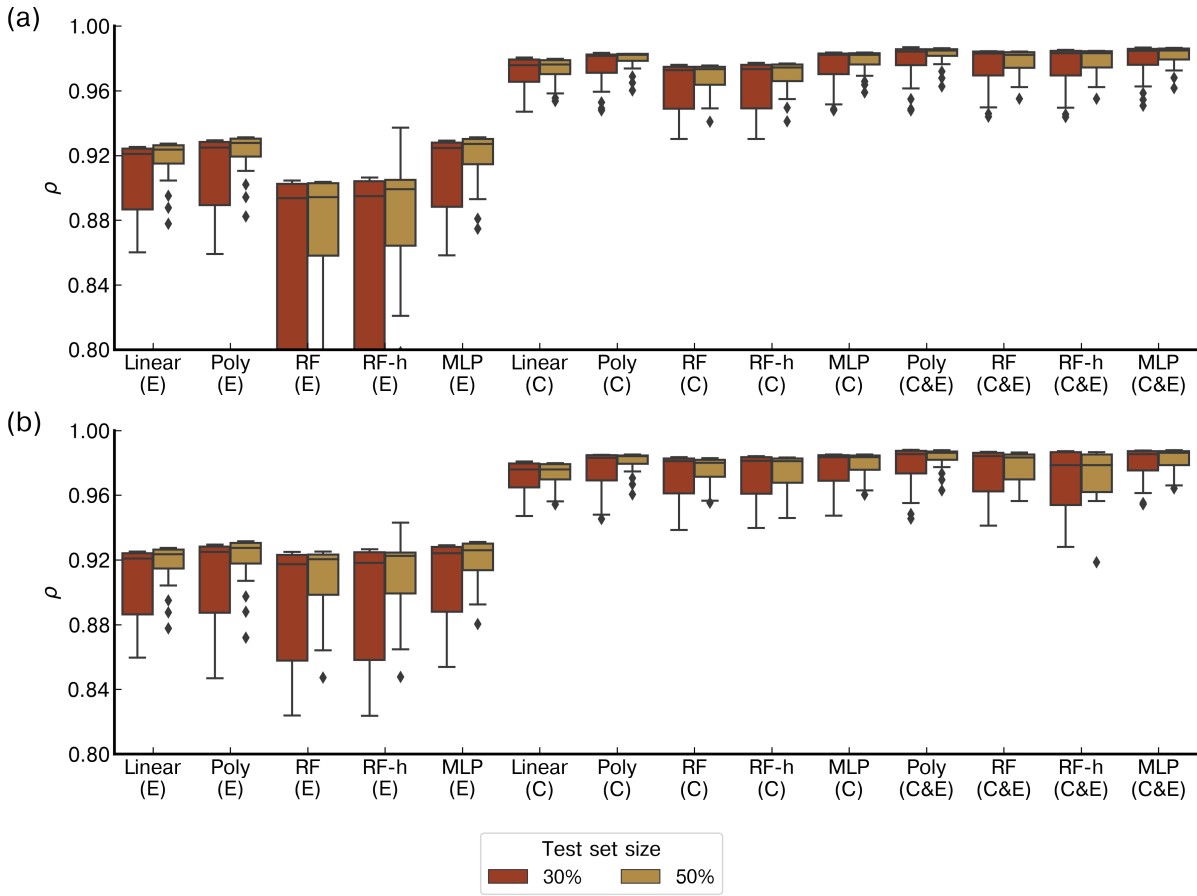

**Figure 6.** Comparison of the Pearson correlation coefficient ($\rho$) distributions between models on the test set for a 20-fold cross validation. The boxes are the inter-quartile of the distribution of $\rho$, the whiskers are the $5^{th}$ and $95^{th}$ percentiles, and the black line is the median (a) Models in which the inputs are only voltage from the Figaro TGS sensors, (b) Models in which the inputs include voltage from low-cost sensors and environmental variables ($H_2O$, Temperature and Pressure). 'Linear' represents the linear or multilinear model, 'Poly' the polynomial model, 'RF' the random forest model, 'RF-h' the random forest hybridized with a polynomial regression, and 'MLP' the multilayer perceptron. Under each model, labels denote which TGS sensor was used; 'C' is the TGS 2611-C00, 'E' the TGS 2611-E00 and 'C & E' both sensors at the same time. The red boxplots represent the results of models with a test set size of 30% of the total observations and the yellow ones a test set size of 50%. Note that the y-axis was limited in a range to distinguish the different models.



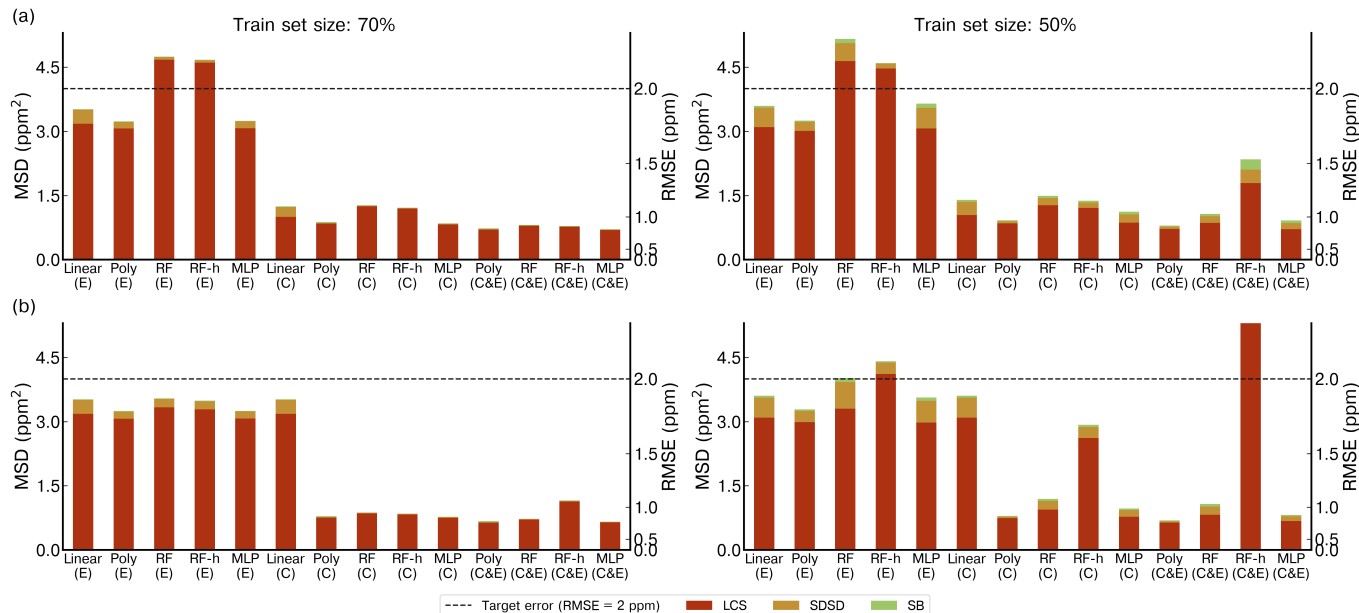

**Figure 7.** Comparison of the Mean standard deviation (MSD) across the different models on the test set for a 20-fold cross validation. (a) Models with only voltage of TGS sensors as input. (b) Models including environmental variables and voltage of TGS sensors in the input. Left panels show the performances on a train set size of 70% and right panels a train set size of 50% of the total observations. The stacked bars show the contribution of each component of the MSD to the total error, the Lack of Positive Correlation weighted by $\sigma$ (LCS) in red, the difference in the magnitude fluctuation (SDSD) in orange and the simulation bias (SB) in green. Notation for the models is the same as for the figure 6.



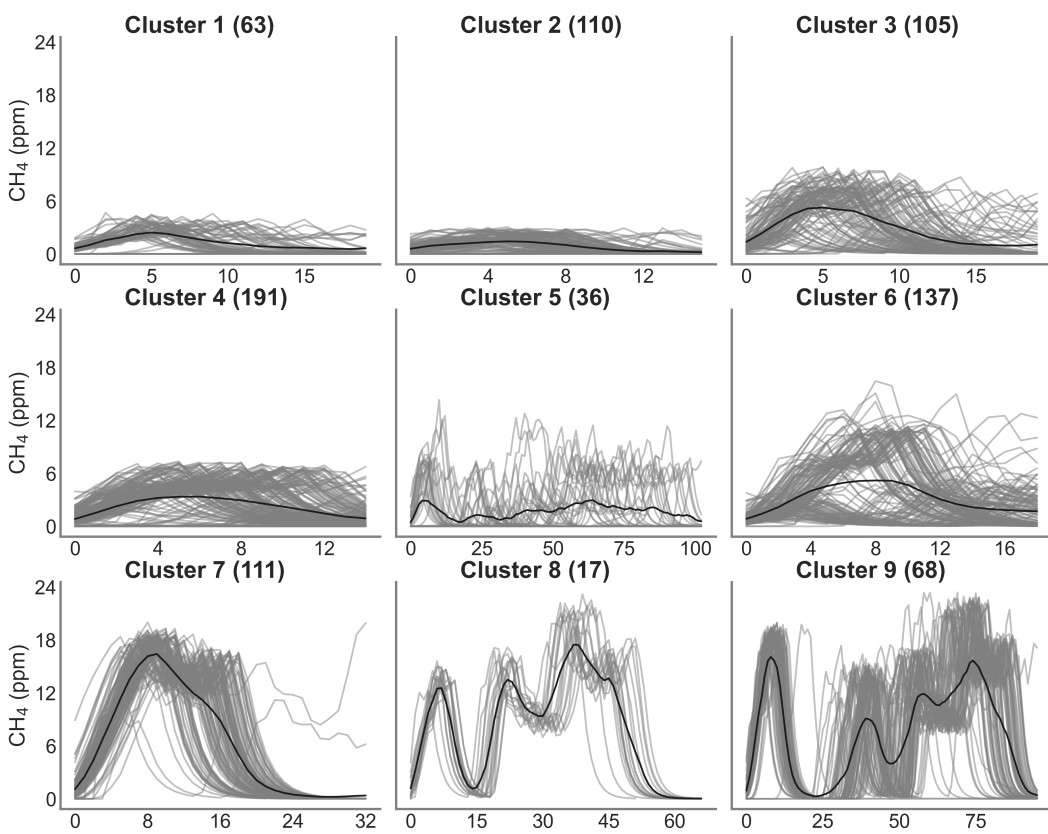

**Figure 8.** Clustering of peaks using DTW on the reference instrument. On the title of each plot the number inside the parentheses corresponds to the number of spikes attributed to each cluster. Thin grey lines represent all the peaks inside each cluster and the black line is the mean of all the peaks corresponding to each class.

**Figure 9.** Performance of each model for the different configurations of training and test set (1 to 11 in the x-axis) considering the identified clusters. (a) Only Figaro TGS 2611-C00 data as input. (b) Only TGS 2611-E00 data as input. (c) Both Figaro sensors data as input. (d) TGS 2611-C00 data and environmental variables. (e) TGS 2611-E00 and environmental variables. (f) Both TGS sensors and environmental variables. Note the different y-axis for the figure (b) and (e).



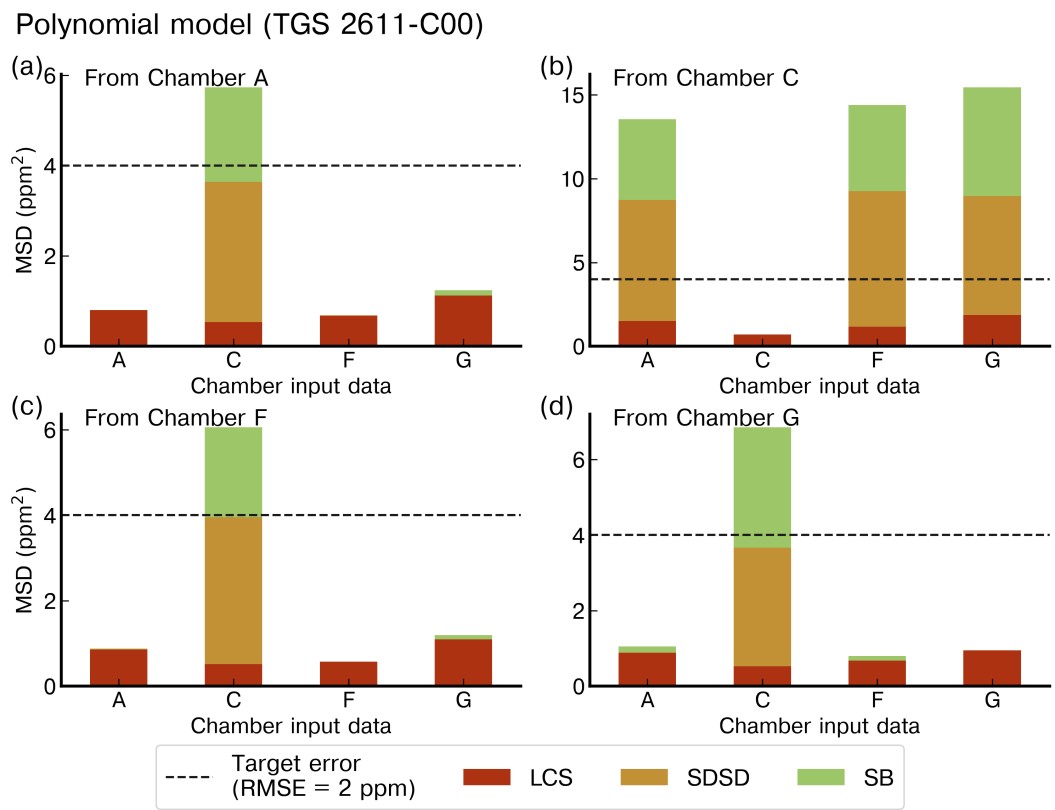

**Figure 10.** Reconstruction error of the peaks for the polynomial model with TGS 2611-C00 as input using the best stratified training case from (a) Chamber A, (b) Chamber C, (c) Chamber F and (d) Chamber G to reconstruct the peaks from the other chambers (listed on the x-axis) with data from the same type of sensor. Note the different ranges of the y-axis for the panels (b) and (c).

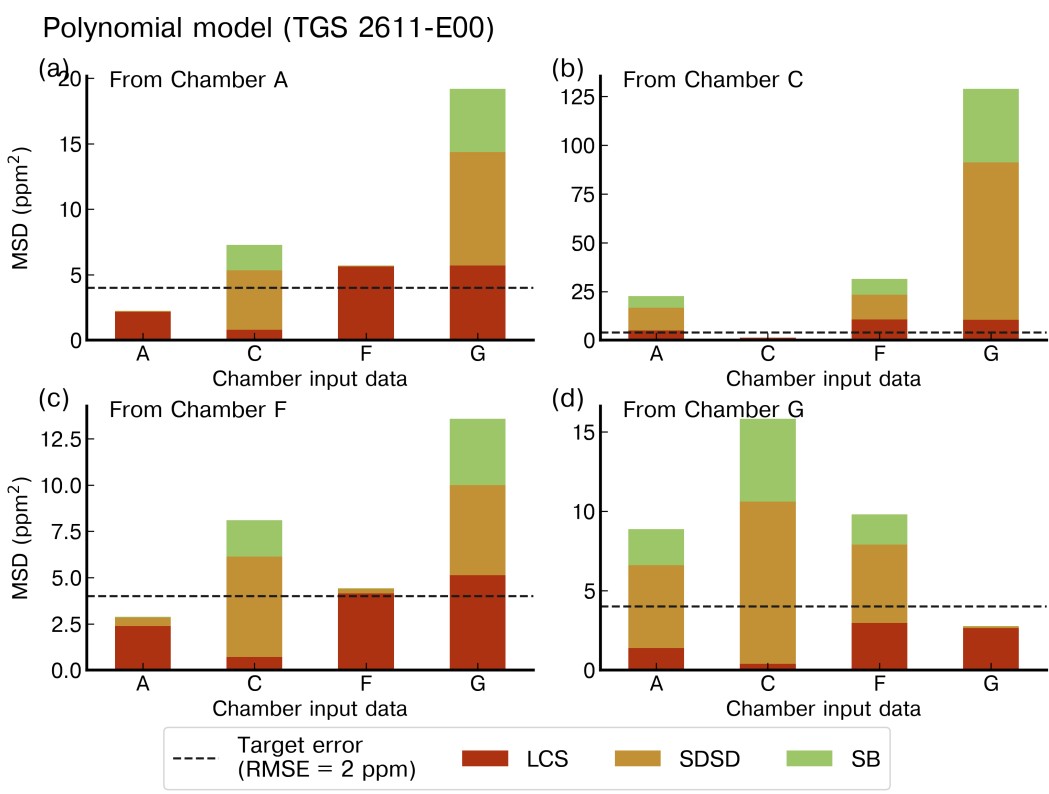

**Figure 11.** Reconstruction error of the peaks for the polynomial model with TGS 2611-E00 as input using the best stratified training case on (a) Chamber A, (b) Chamber C, (c) Chamber F and (d) Chamber G to reconstruct the peaks from the other chambers with data from the same type of sensor. Note the different ranges of the y-axis.

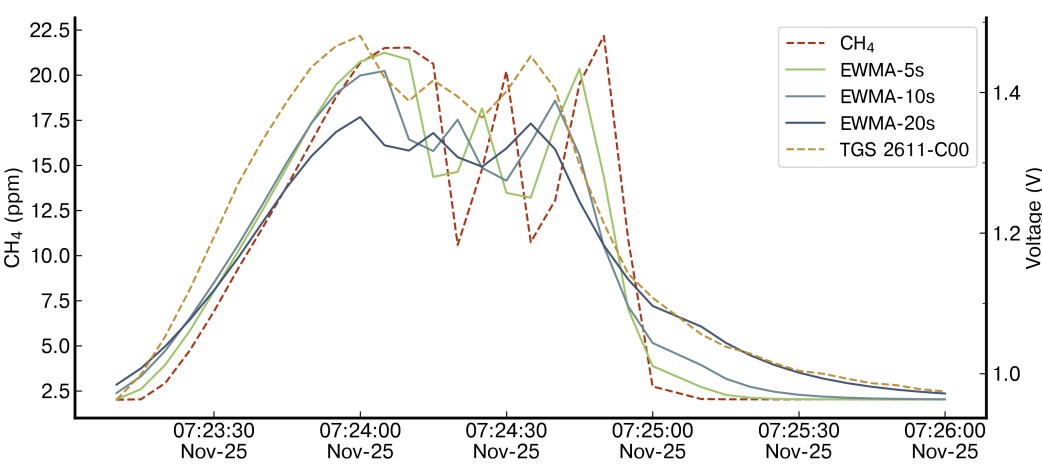

**Figure A1.** Different time constant values of the exponential weighted moving average (EWMA) applied to the reference instrument. The reference instrument in red dotted lines, the applied smoothing for three values of time constant (5s, 10s, and 20s) denoted 'EWMA' for one peak and the TGS 2611-C00 voltage from logger A to compare the smoothing effect in yellow dotted.



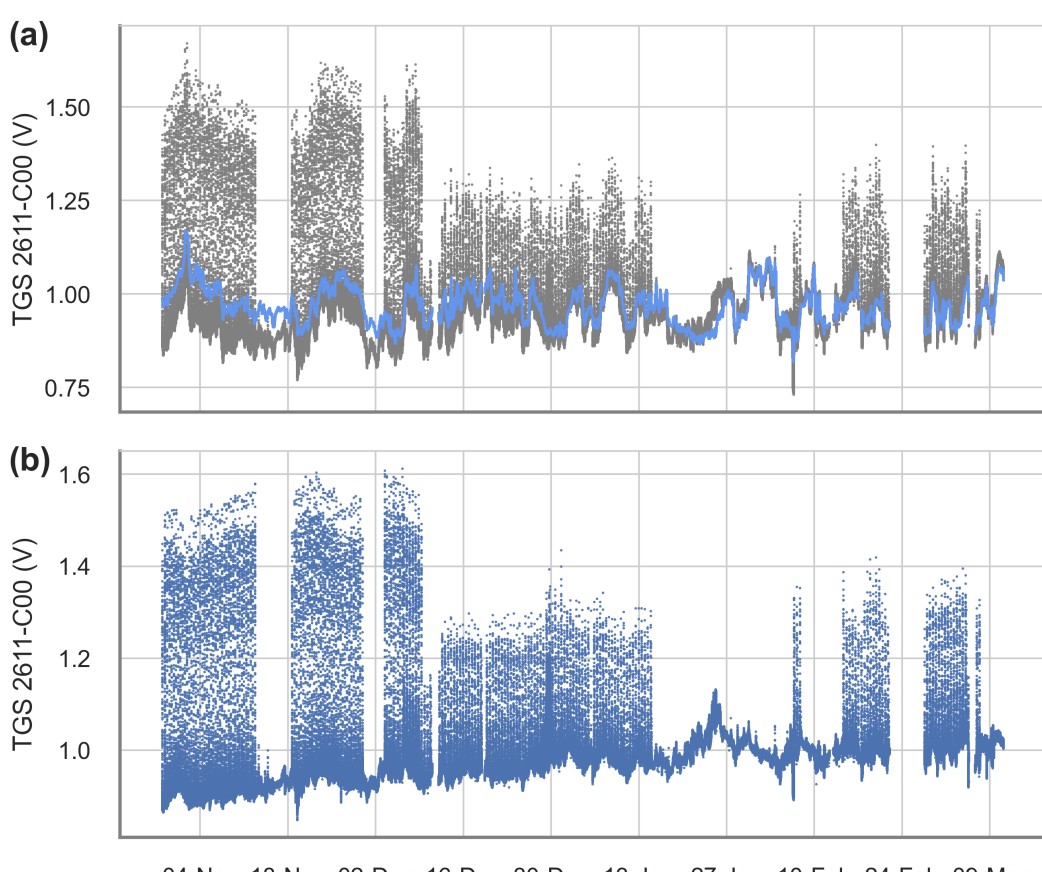

**Figure A2.** Derived contribution and correction of water vapor for the Figaro TGS 2611-C00. (a) The raw voltage signal (gray) and the derived cross-sensitivities to $H_2O$ (blue). (b) The cross-sensitivity corrected signal.



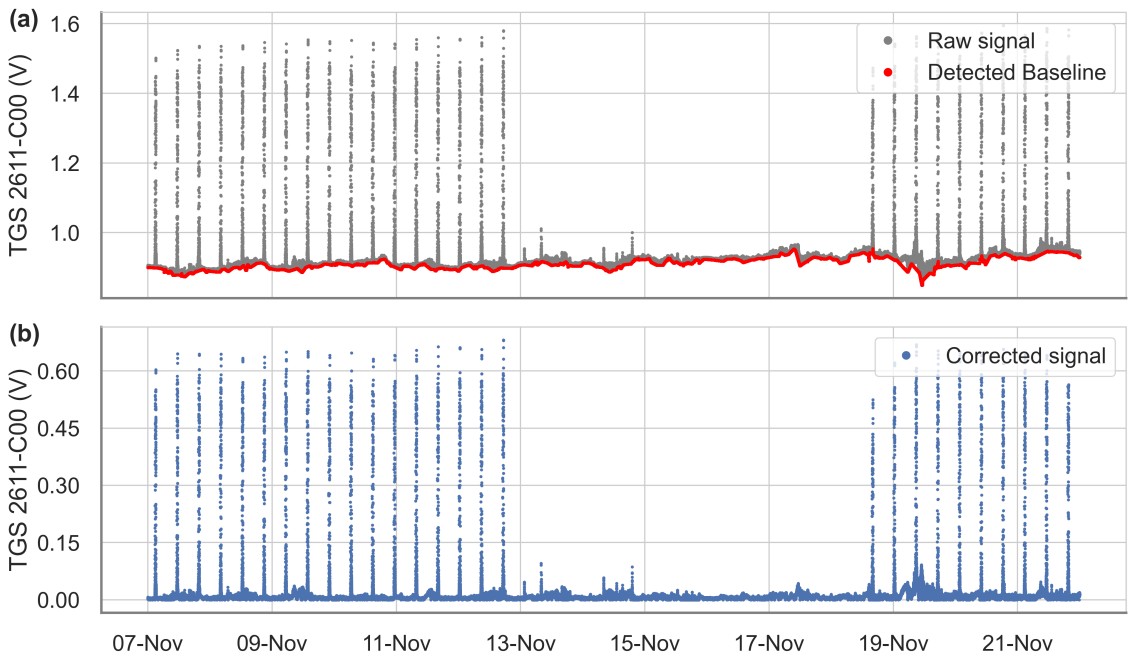

**Figure A3.** Example of the baseline extraction and correction for the Figaro TGS 2611-C00 over 15 days. (a) Raw signal (gray) and detected baseline with the spike detection algorithm (red). (b) Voltage signal with the corrected baseline.





**Figure A4.** Time series of the reference CH$_4$ signal, Figaro® TGS sensor, and environmental variables for the entire experiment.

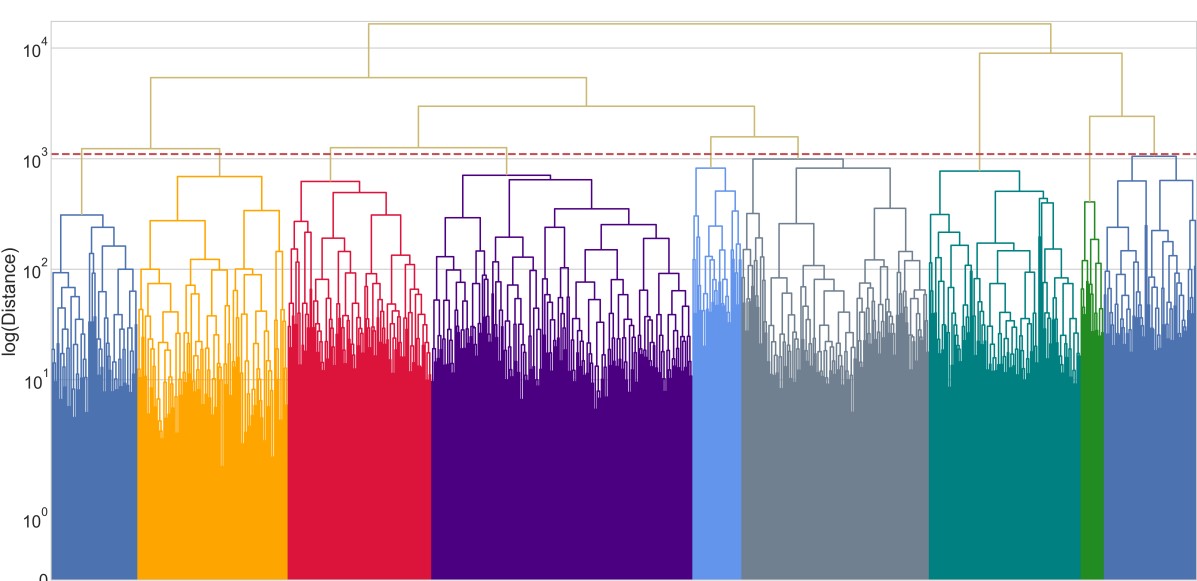

**Figure A5.** Dendrogram constructed from the distance matrix computed using the DWT metric. Red dotted line represents the threshold used to determine the clusters. Each color under the threshold line represents one cluster of peaks. Note that y-axis was rescaled to the logarithm of the 'ward' distance to appreciate better the threshold and the clusters.



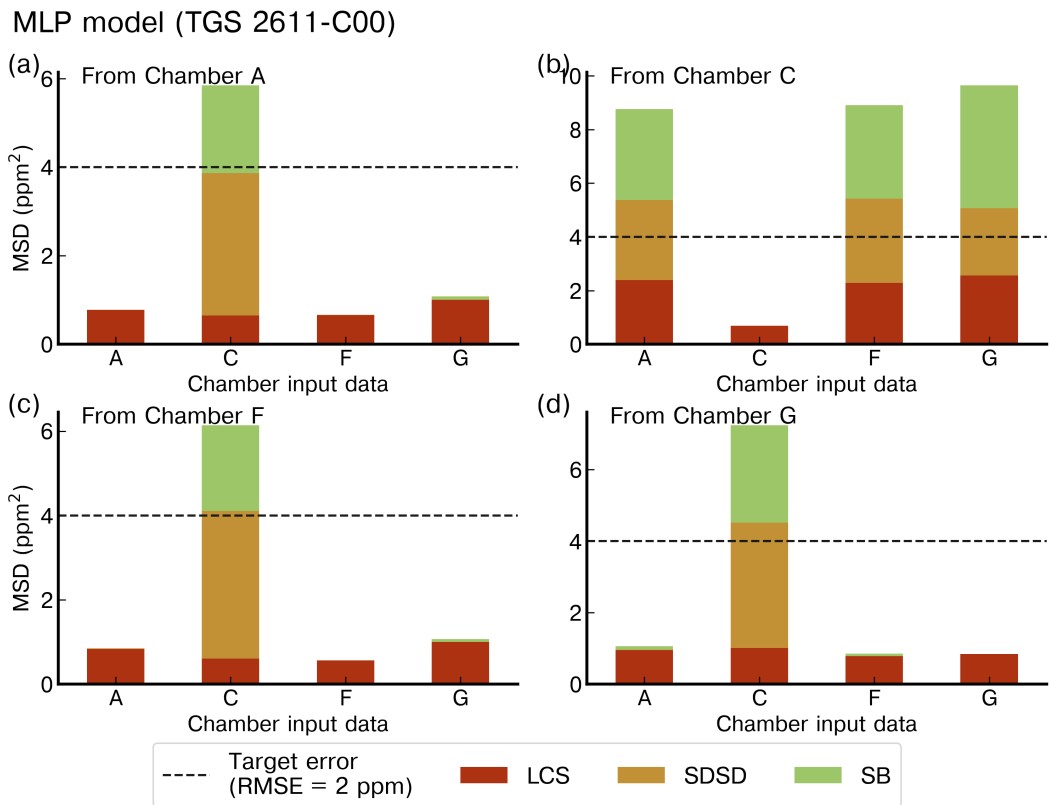

**Figure A6.** Reconstruction error of the peaks for the MLP model with the TGS 2611-C00 as input and using the best stratified training on the (a) Chamber A, (b) Chamber C, (c) Chamber F and (d) Chamber G. The first column on each panel is the reconstruction error on the test set of the chamber on which the training was made, the other columns are the reconstruction on the whole dataset for that chamber on the same sensor. Note the different ranges of the y-axis.



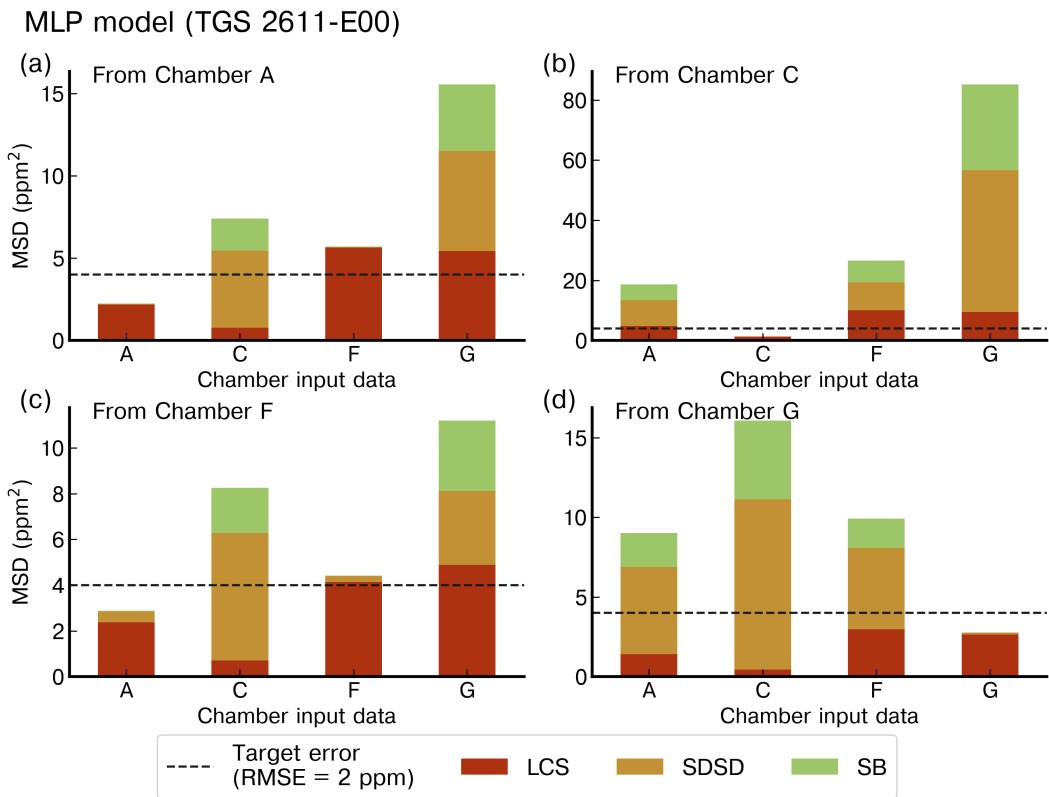

**Figure A7.** Reconstruction error of the peaks for the MLP model with the TGS 2611-E00 as input and using the best stratified training on the (a) Chamber A, (b) Chamber C, (c) Chamber F and (d) Chamber G. The first column on each panel is the reconstruction error on the test set of the chamber on which the training was made, the other columns are the reconstruction on the whole dataset for that chamber on the same sensor. Note the different ranges of the y-axis.



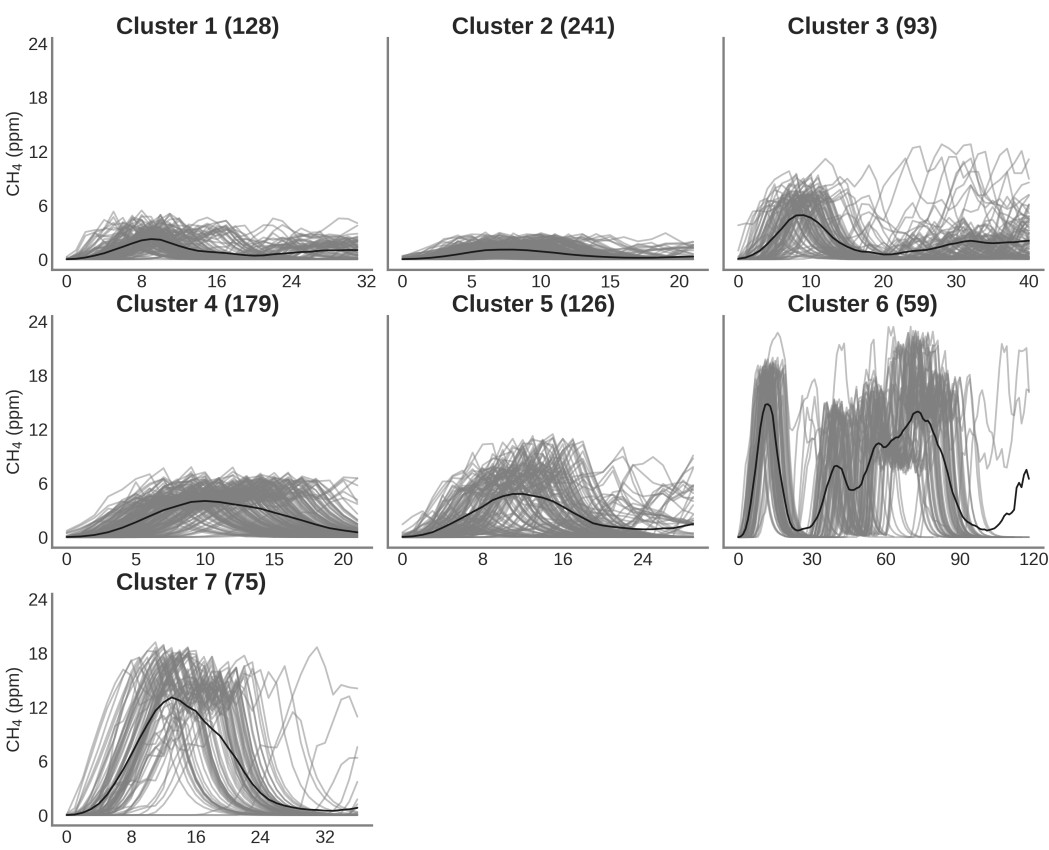

**Figure A8.** Clustering of peaks using DTW on the reference instrument for the same spikes detected by sensors on Chamber C. On the title of each plot the number inside the parentheses correspond to the number of spikes attributed to each cluster. Thin gray lines represent all the peaks inside each cluster and the black line is the mean of all the peaks corresponding to each class.



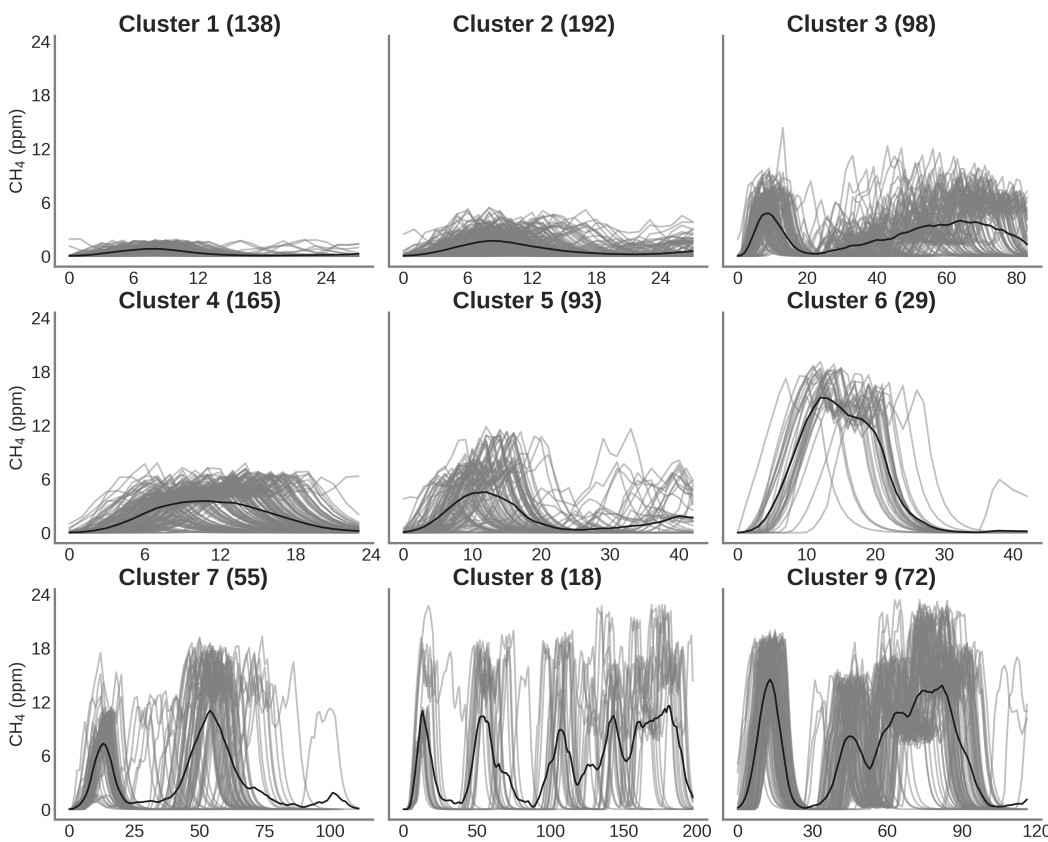

**Figure A9.** Clustering of peaks using DTW on the reference instrument for the same spikes detected by sensors on Chamber F. On the title of each plot the number inside the parentheses correspond to the number of spikes attributed to each cluster. Thin gray lines represent all the peaks inside each cluster and the black line is the mean of all the peaks corresponding to each class.



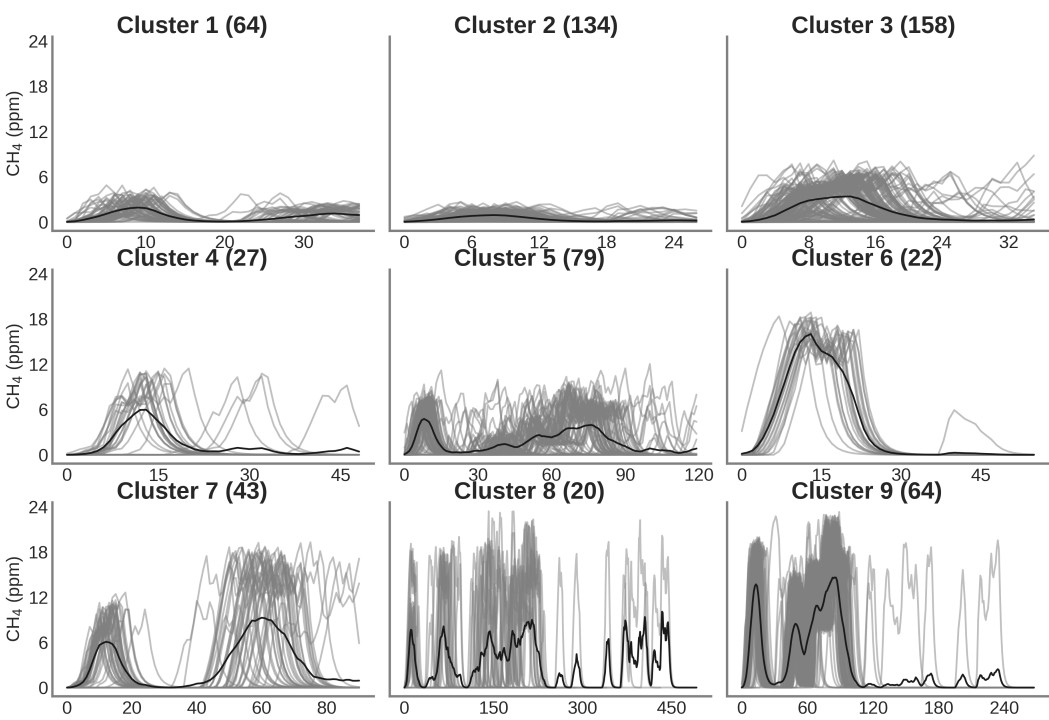

**Figure A10.** Clustering of peaks using DTW on the reference instrument for the same spikes detected by sensors on Chamber G. On the title of each plot the number inside the parentheses correspond to the number of spikes attributed to each cluster. Thin gray lines represent all the peaks inside each cluster and the black line is the mean of all the peaks corresponding to each class.





**Table A1.** MSD decomposition for the different configurations and both test set sizes considering only Figaro® TGS sensors as input. Letters inside parentheses indicates the sensor used: TGS 2611-E00 denoted 'E', TGS2611-C00 denoted 'C' and both sensors as input denotes 'C&E'

| Test set size | Model | MSD ($ppm^2$) | LCS ($ppm^2$) | SDSD ($ppm^2$) | SB ($ppm^2$) |
|---|---|---|---|---|---|
| 30% | Linear (E) | 3.51 | 3.17 | 0.326 | 0.013 |
| | Poly (E) | 3.23 | 3.06 | 0.155 | 0.014 |
| | RF (E) | 4.74 | 4.67 | 0.061 | 0.011 |
| | RF-h (E) | 4.67 | 4.60 | 0.057 | 0.010 |
| | MLP (E) | 3.24 | 3.07 | 0.163 | 0.011 |
| | Linear (C) | 1.24 | 1.00 | 0.229 | 0.015 |
| | Poly (C) | 0.88 | 0.84 | 0.016 | 0.021 |
| | RF (C) | 1.27 | 1.24 | 0.014 | 0.012 |
| | RF-h (C) | 1.21 | 1.19 | 0.012 | 0.011 |
| | MLP (C) | 0.85 | 0.82 | 0.014 | 0.014 |
| | Poly (C&E) | 0.73 | 0.70 | 0.010 | 0.023 |
| | RF (C&E) | 0.81 | 0.79 | 0.009 | 0.009 |
| | RF-h (C&E) | 0.78 | 0.76 | 0.008 | 0.009 |
| | MLP (C&E) | 0.71 | 0.69 | 0.009 | 0.011 |
| 50% | Linear (E) | 3.59 | 3.09 | 0.457 | 0.042 |
| | Poly (E) | 3.24 | 3.00 | 0.218 | 0.022 |
| | RF (E) | 5.16 | 4.64 | 0.415 | 0.103 |
| | RF-h (E) | 4.59 | 4.46 | 0.115 | 0.014 |
| | MLP (E) | 3.64 | 3.06 | 0.481 | 0.097 |
| | Linear (C) | 1.39 | 1.04 | 0.304 | 0.050 |
| | Poly (C) | 0.92 | 0.84 | 0.047 | 0.025 |
| | RF (C) | 1.49 | 1.27 | 0.164 | 0.054 |
| | RF-h (C) | 1.37 | 1.20 | 0.127 | 0.040 |
| | MLP (C) | 1.11 | 0.86 | 0.192 | 0.063 |
| | Poly (C&E) | 0.79 | 0.71 | 0.051 | 0.030 |
| | RF (C&E) | 1.07 | 0.85 | 0.158 | 0.054 |
| | RF-h (C&E) | 2.34 | 1.78 | 0.314 | 0.243 |
| | MLP (C&E) | 0.91 | 0.71 | 0.144 | 0.059 |



**Table A2.** MSD decomposition for the different configurations and both test set sizes considering Figaro® TGS sensors and environmental variables as input. Notation is the same as on Table A1

| Test set size | Model | MSD (ppm$^2$) | LCS (ppm$^2$) | SDSD (ppm$^2$) | SB (ppm$^2$) |
|---|---|---|---|---|---|
| 30% | Linear (E) | 3.59 | 3.09 | 0.45 | 0.042 |
| | Poly (E) | 3.24 | 3.00 | 0.21 | 0.022 |
| | RF (E) | 5.16 | 4.64 | 0.41 | 0.103 |
| | RF-h (E) | 4.59 | 4.46 | 0.11 | 0.014 |
| | MLP (E) | 3.64 | 3.07 | 0.48 | 0.097 |
| | Linear (C) | 1.39 | 1.04 | 0.30 | 0.050 |
| | Poly (C) | 0.92 | 0.84 | 0.04 | 0.025 |
| | RF (C) | 1.49 | 1.27 | 0.16 | 0.054 |
| | RF-h (C) | 1.37 | 1.20 | 0.12 | 0.040 |
| | MLP (C) | 1.11 | 0.86 | 0.19 | 0.063 |
| | Poly (C&E) | 0.79 | 0.71 | 0.05 | 0.030 |
| | RF (C&E) | 1.07 | 0.85 | 0.15 | 0.054 |
| | RF-h (C&E) | 2.34 | 1.78 | 0.31 | 0.243 |
| | MLP (C&E) | 0.91 | 0.71 | 0.14 | 0.059 |
| 50% | Linear (E) | 3.60 | 3.09 | 0.46 | 0.045 |
| | Poly (E) | 3.28 | 2.98 | 0.26 | 0.034 |
| | RF (E) | 4.02 | 3.30 | 0.62 | 0.094 |
| | RF-h (E) | 4.41 | 4.11 | 0.26 | 0.032 |
| | MLP (E) | 3.56 | 2.97 | 0.50 | 0.077 |
| | Linear (C) | 3.60 | 3.09 | 0.46 | 0.045 |
| | Poly (C) | 0.79 | 0.74 | 0.03 | 0.020 |
| | RF (C) | 1.19 | 0.94 | 0.20 | 0.051 |
| | RF-h (C) | 2.92 | 2.61 | 0.25 | 0.049 |
| | MLP (C) | 0.96 | 0.77 | 0.15 | 0.035 |
| | Poly (C&E) | 0.69 | 0.64 | 0.02 | 0.023 |
| | RF (C&E) | 1.07 | 0.82 | 0.19 | 0.056 |
| | RF-h (C&E) | 33933.88 | 566.99 | 33319.58 | 47.296 |
| | MLP (C&E) | 0.81 | 0.67 | 0.11 | 0.031 |





**Table A3.** RMSE in ppm for the different configurations of subsetting based on the selected clusters of peaks. Only Figaro® sensors were used to compute this errors. Each configuration is denoted 'CX' with X the number of the configuration. On each row the models are denoted with a letter inside parentheses to indicate the sensors used. 'C' for the TGS 2611-C00, 'E' for the TGS 2611-E00 and 'C&E' for both sensors.

| | C1 | C2 | C3 | C4 | C5 | C6 | C7 | C8 | C9 | C10 | C11 |
|---|---|---|---|---|---|---|---|---|---|---|---|
| Linear (C) | 1.121 | 1.115 | 1.126 | 1.119 | 1.163 | 1.141 | 1.142 | 1.105 | 1.098 | 1.074 | 1.105 |
| Poly (C) | 0.962 | 0.966 | 0.972 | 0.963 | 0.988 | 0.982 | 0.970 | 0.943 | 0.952 | 0.902 | 0.957 |
| RF (C) | 1.140 | 1.135 | 1.149 | 1.137 | 1.176 | 1.158 | 1.136 | 1.111 | 1.123 | 1.075 | 1.128 |
| RF-h (C) | 1.118 | 1.113 | 1.127 | 1.113 | 1.152 | 1.136 | 1.112 | 1.088 | 1.101 | 1.061 | 1.106 |
| MLP (C) | 0.943 | 0.951 | 0.957 | 0.943 | 0.976 | 0.964 | 0.948 | 0.926 | 0.928 | 0.893 | 0.937 |
| Linear (E) | 1.971 | 1.932 | 1.951 | 1.926 | 1.988 | 1.972 | 1.947 | 1.838 | 1.887 | 1.772 | 1.909 |
| Poly (E) | 1.899 | 1.861 | 1.874 | 1.852 | 1.909 | 1.896 | 1.870 | 1.764 | 1.817 | 1.703 | 1.838 |
| RF (E) | 2.277 | 2.229 | 2.261 | 2.207 | 2.261 | 2.251 | 2.184 | 2.264 | 2.233 | 2.230 | 2.235 |
| RF-h (E) | 2.263 | 2.214 | 2.245 | 2.192 | 2.243 | 2.235 | 2.165 | 2.249 | 2.218 | 2.221 | 2.219 |
| MLP (E) | 1.898 | 1.861 | 1.874 | 1.853 | 1.910 | 1.895 | 1.869 | 1.764 | 1.816 | 1.705 | 1.838 |
| Linear (C&E) | 1.121 | 1.115 | 1.126 | 1.121 | 1.164 | 1.142 | 1.144 | 1.105 | 1.098 | 1.074 | 1.105 |
| Poly (C&E) | 0.872 | 0.890 | 0.895 | 0.885 | 0.904 | 0.911 | 0.891 | 0.869 | 0.881 | 0.844 | 0.886 |
| RF (C&E) | 0.887 | 0.943 | 0.951 | 0.931 | 0.978 | 0.936 | 0.938 | 0.919 | 0.936 | 0.914 | 0.938 |
| RF-h (C&E) | 0.878 | 0.929 | 0.940 | 0.917 | 0.964 | 0.925 | 0.927 | 0.906 | 0.924 | 0.907 | 0.924 |
| MLP (C&E) | 0.861 | 0.948 | 0.875 | 0.874 | 0.970 | 0.958 | 0.871 | 0.841 | 0.862 | 0.820 | 0.935 |





**Table A4.** RMSE in ppm for the different configurations of subsetting based on the selected clusters of peaks. Figaro® sensors and environmental variables were used to compute this errors. Notation is the same as on Table A3.

| | C1 | C2 | C3 | C4 | C5 | C6 | C7 | C8 | C9 | C10 | C11 |
|---|---|---|---|---|---|---|---|---|---|---|---|
| Linear (C) | 1.115 | 1.111 | 1.118 | 1.112 | 1.155 | 1.143 | 1.136 | 1.106 | 1.095 | 1.075 | 1.101 |
| Poly (C) | 0.915 | 0.920 | 0.928 | 0.913 | 0.928 | 0.935 | 0.919 | 0.887 | 0.898 | 0.872 | 0.912 |
| RF (C) | 0.941 | 0.988 | 0.997 | 0.980 | 1.010 | 0.988 | 0.967 | 0.937 | 0.950 | 0.905 | 0.977 |
| RF-h (C) | 0.925 | 2721.300 | 2738.400 | 2752.600 | 2842.700 | 2800.200 | 2827.700 | 2788.600 | 0.949 | 0.902 | 2681.000 |
| MLP (C) | 0.899 | 0.918 | 0.914 | 0.905 | 0.944 | 0.926 | 0.905 | 0.872 | 0.884 | 0.854 | 0.894 |
| Linear (E) | 1.971 | 1.932 | 1.951 | 1.926 | 1.988 | 1.972 | 1.948 | 1.839 | 1.888 | 1.773 | 1.910 |
| Poly (E) | 1.897 | 1.863 | 1.879 | 1.856 | 1.910 | 1.897 | 1.874 | 1.763 | 1.820 | 1.704 | 1.842 |
| RF (E) | 1.979 | 1.963 | 1.978 | 1.951 | 2.003 | 2.000 | 1.959 | 1.874 | 1.939 | 1.834 | 1.951 |
| RF-h (E) | 1.969 | 1370.000 | 13792.900 | 13864.300 | 14317.900 | 14103.900 | 14242.700 | 14045.700 | 1.932 | 1.830 | 13506.000 |
| MLP (E) | 1.897 | 1.865 | 1.879 | 1.860 | 1.914 | 1.898 | 1.875 | 1.801 | 1.826 | 1.706 | 1.844 |
| Linear (C&E) | 1.115 | 1.111 | 1.118 | 1.114 | 1.156 | 1.143 | 1.139 | 1.106 | 1.095 | 1.076 | 1.101 |
| Poly (C&E) | 0.833 | 0.851 | 0.858 | 0.842 | 0.851 | 0.864 | 0.850 | 0.824 | 0.836 | 0.828 | 0.849 |
| RF (C&E) | 0.817 | 0.886 | 0.891 | 0.888 | 0.920 | 0.864 | 0.874 | 0.855 | 0.867 | 0.847 | 0.885 |
| RF-h (C&E) | 0.811 | 0.914 | 0.924 | 0.925 | 0.951 | 0.902 | 0.911 | 0.893 | 0.893 | 0.856 | 0.913 |
| MLP (C&E) | 0.833 | 0.858 | 0.880 | 0.850 | 0.864 | 0.883 | 0.857 | 0.814 | 1.524 | 0.811 | 0.848 |