# Peer review of "Reconstruction of high-frequency methane atmospheric concentration peaks from measurements using metal oxide low-cost sensors"

_Atmospheric Measurement Techniques, 2022_

## Author Response (AR1)

**Author's response**

**Comments from Referees**

Author's response

Author's changes in manuscript

1. **RC1: It seems that the parsimonious tests could explore the minimum number of peaks required for training on each cluster to achieve satisfiable performance. It is also possible that the polynomial regression may need fewer cases to train than the multilayer perceptron model to achieve the same performance and knowing how many peaks are needed for training would help inform a cost-effective strategy for deployment.**

   The parsimonious test conducted have explored the relative distribution of peaks required to have a good reconstruction of the spikes with a limited dataset. We consider that this approach will be more interesting since we explored several configurations of training sets with relative distributions of the peaks inside each configuration. For a cost effective deployment strategy, it will be necessary have a better understanding of other factors, like cross-sensitivity to other gases, before develop a more in-dept analysis of the number of spikes required to train the models.

   We acknowledge this comment. We included a statement in the conclusion section:

   Understanding the number of spikes required for the training of the models can help to define a cost-effective strategy for deployment of the sensors.

2. **RC1: Another issue is with the cross-sensitivity. Because it is stated that MOS sensors are sensitive to electron donors other than CH4, I wonder if the presence of ethane in natural gas would cause a problem. This potential limitation would need to be accounted for or acknowledged at least.**

   The principal idea of the experiment was to characterize the capabilities of TGS sensors to measure enhancements of $CH_4$ similar to typical $CH_4$ leaks found on oil and gas facilities. The experiment was designed to expose the sensors in a controlled environment where we can simulate the spikes of $CH_4$ without other interfering gases. The next step following this

study will be an extensive characterization of interfering gases present on gas production facilities. We acknowledge this comment, so we propose a statement in the discussion section:

The presence of other electron donors, such as ethane, isobutane, etc., also needs to be accounted in the model as a predictor or in the correction of the baseline.

3. **RC1: I do not see a Data Availability section, and I suggest the authors check if they conform with the journal's data policy.**

   A statement was added on the manuscript.

   The dataset was collected and the codes developed in the frame of the Chaire Indutrielle Trace ANR-17-CHIN-0004-01. They are accessible upon request to the corresponding author.

4. **RC1: The writing is overall quite clear, but some grammatical errors and typos need to be fixed.**

   The writing of the manuscript was reviewed to correct the grammatical errors and typos.

5. **RC1: L10–11: "The obtained relative accuracy is higher than 10% to reconstruct the maximum amplitude of peaks (RMSE ≤ 2 ppm)" - There is ambiguity in "higher accuracy" - does it mean that the RMSE is lower than 10% of the peak amplitude? If so, it is better to say that the relative accuracy is better than 10%.**

   The reviewer is right regarding the meaning of this sentence. We agree with his suggestion for the rewriting. The statement in the abstract was corrected.

   The obtained relative accuracy is better than 10% to reconstruct the maximum amplitude of peaks (RMSE ≤ 2 ppm)

6. **RC1: L24: "Anthropogenic CH4 emissions account for 60% of global emissions (Saunois et al., 2016)" - This figure may be updated with the latest Global Methane Budget estimates (Saunois et al., 2020, ESSD, https://doi.org/10.5194/essd-12-1561-2020).**

   We have corrected the reference to Saunois et al, (2020)

   Anthropogenic $CH_4$ emissions account for 60% of global emissions (Saunois et al., 2020).

7. **RC1: L60: "based on the observed voltage of each sensor and other variables" - What are the other variables?**

   The sentence was corrected. Here is the new version of the sentence:

This study aims to compare several parametric (linear and polynomial) and non-parametric models (random forest, hybrid random forest and ANN) applied to different combinations of Figaro TGS sensors to reconstruct the CH4 signals of repeated atmospheric spikes, based on the observed voltage of each sensor. In addition, environmental variables measured from other low cost-sensors in parallel to TGS sensors, such as air temperature and pressure and $H_2O$ mole fraction, were also added as predictors to the models.

8. **RC1: L63: How would you expect to capture a spike of "several tenths of ppm" above the background (Kumar et al., 2021) using a sensor with accuracy no better than 0.8 ppm?**

   The sentence was corrected. Here is the new version of the sentence:

   The CH4 signal we aim to reconstruct is representative of variations observed in the atmosphere from leaks that occur within or close to an emitting industrial facility, i.e. short duration CH4 enhancements (spikes) lasting between 1 to 7 minutes and ranging from few tenth of ppm to few ppm above an atmospheric background concentration of around 2 ppm (Kumar et al., 2021)

9. **RC1: L101: 2.1.1 describes only five of the six chambers. Table 1: Why is Chamber B excluded?**

   There is indeed an error on the sentence, we have only installed 5 chambers, but we had initially previously assembled 6 chambers. One of the chamber (Chamber B) had issues with the logging system and thus was removed from the study. We have corrected the naming of the chambers (from A to E) to prevent further confusion and the text was updated accordingly.

10. **RC1: L132: Is there a compelling reason to down sample the data to 5 s resolution instead of 2 s?**

    There is indeed an error on the sentence, we have only installed 5 chambers, but we had initially previously assembled 6 chambers. One of the chamber (Chamber B) had issues with the logging system and thus was removed from the study. We have corrected the naming of the chambers (from A to E) to prevent further confusion and the text was updated accordingly.

11. **RC1: L153: Does β represent 3.5 ppm or 3.5 standard deviations?**

    β is the number of standard deviations. The text was corrected.

The threshold or scale parameter, β, defines a range in number of standard deviations around the modelled baseline. A value of β = 3.5 was used.

12. **RC1: L273–275: This sentence seems to belong to the methods.**

The sentence has been moved from results section (3.1 Data pre-processing and baseline correction) to the method section (2.1.2 Generation of methane spikes on top of ambient air)

13. **RC1: L288 and Fig. 5: It appears that the peaks measured by the Type E sensor lag behind those measured by the CRDS. Has the time lag been accounted for properly? Why do the peaks measured by the Type E sensor appear more dampened than those measured by the Type C sensor when both were in the same chamber?**

Yes, signals were aligned properly. It was also accounted in the correction the time lag of 10 s after applying the EWMA on the CRDS. The time lag observed on the Type E sensor is due to the carbon filter added on top of the sensing material to improve the selectivity to $CH_4$ on the sensor. This carbon filter also produces an airflow resistance leading to a slower response.

A statement was included on the manuscript:

This behavior can be linked to the carbon filter included on top of the sensing material of type E sensor that produces an airflow resistance leading to a slower response.

14. **RC1: L301: "interquartile range (IQ) = 0.001" - The interquartile ranges presented in Fig. 6 seem substantially larger than 0.001.**

The text mentioning the interquartile ranges were corrected.

Among the models in the first group, the Polynomial Model gave the largest correlations ($\rho_{Median}$ = 0.92, interquartile range (IQ) = 0.035 and 0.013 for a test set size of 30% and 50% respectively).

15. **RC1: L303: Again, check the interquartile ranges. Unless I'm misreading Fig. 6, the interquartile ranges seem substantially larger than indicated here.**

The IQ ranges were corrected.

Of note is that 2 tables have been added in the supplementary material (Table A5 and A6). These two tables show summary statistics: correlation coefficients distribution of the 20-fold cross validation.

[revised manuscript text omitted]

16. **RC1: Figs. 7 and 9: Remove the axis on the right-hand side of each panel; it's unnecessary and potentially confusing. Instead, indicate that the gray dashed lines represent the target accuracy of RMSE = 2 ppm or MSD = 4 ppm².**

    Figures were updated following this suggestion.

17. **RC1: L362–363: "We observed that after six months, the RMSE error produced by the models increased from 0.57 to 0.85 ppm." - This sentence is confusing. I thought the RMSE from the first experiment was 0.57 ppm for a moment, but it turned out to be the difference in RMSE. Please rewrite to clarify.**

    The sentence was replaced by this new one:

    We observed that after six months, the RMSE had increased in a range of 0.57-0.85 ppm on the second experiment.

18. **RC1: L397: "poorest" -> "poorer" - You are only comparing two sensors.**

    Accepted suggestion. The manuscript was updated.

    The TGS 2611-E00 (Type E) was the sensor with the poorer performance, regardless of the model employed, or of the subset of data used to train models, as shown by our tests with 5 chambers, each containing 5 different sensors.

19. **RC1: L401: Does the carbon filter create a barrier to diffusion?**

    Yes, as it is stated in the "Technical Application Notes of TGS 2611 sensors" provided by the manufacturer: "the filter present on top of the sensing material of TGS 2611-E00 add an airflow resistance producing a slower response".

20. **RC1: L428–429: "... an RMSE of the residuals of 0.043 μmol mol−1 (0.69 ppm)" - This statement doesn't make sense, because μmol mol–1 and ppm are the same units, unless by ppm you mean something different from the volume fraction.**

We acknowledge the correction. The statement was updated.

Their overall model performances for seven years of continuous $CH_4$ monitoring on ambient air in northern Alaska (range of variation between 1.7 and 2.1 ppm) with a Figaro TGS2600 gave an RMSE of the residuals of 0.043 $\mu mol\ mol^{-1}$.

21. **RC1: L439–440: "we were able to reduce the length of the training dataset from 70% to 25% while maintaining similar performance" - But the caveat here is that you need to use 70% of the data for a certain cluster and 10% of the data for all the rest of clusters to achieve optimal performance (25% of all peaks). Without a careful characterization of the diversity of spike shapes, we won't be able to know which cluster(s) to prioritize when collecting training data**

Yes, the proposed methodology accounts for an *a priori* knowledge of the typical spikes to which the sensors will be exposed. However, in our dataset, the cluster that provides best performances in the reconstruction corresponds to spikes that covers a wide range of concentrations. This shows that models benefit from having a subset of training data containing a wide diversity of examples (here in the form of concentrations since we don't include time relationships in the models).

A statement is added in the discussion section:

This approach is designed with an *a priori* knowledge of the typical concentrations the sensors will be exposed. Although, exposing to a wide range of concentrations, like the ones included on cluster 9 from our experiment, can lead to have a large variety of examples for the training of the calibration models.

22. **RC2: L23-24: The statements read as if natural gas accounts for all the anthropogenic $CH_4$**
A statement was added:
Emissions from natural gas production accounts for 63% of the total emissions from fossil fuel production and use (Saunois et al., 2020).

23. **RC2: L65-66: How could the influence of other VOC on the measurement be addressed?**
A statement was added:
The main focus of this study is the behavior of TGS sensors that are exposed to enhancements of $CH_4$ on top of background signal without the presence of other interfering gases. The influence of VOCs on a real deployment should be considered and included as a predictor to

the reconstruction models, corrected on a preprocessing stage by determining the sensitivity of TGS to them or determine, from specific laboratory experiments, the amount of signal that models can filter out and the needs in terms of ancillary measurements.

24. **RC2: L89-90: It would be worth mentioning here how the three Figaro TGS sensors differ from each other, and why only Types C and E were used for the analysis.**

A table was included in the supplementary material and the text was updated to explain the differences between the sensors:

Each chamber contained a Figaro TGS 2600 originally designed to measure VOCs but sensitive to $CH_4$, TGS 2611-C00 with enhanced sensitivity to $CH_4$ and TGS 2611-E00 that includes a carbon filter on top of the sensing material to improve the selectivity to $CH_4$ even further (see Table A7 for information on the differences of each TGS sensor), alongside a relative humidity and temperature sensor (DHT22 or Sensirion SHT75), and a temperature and pressure sensor (Bosch BMP280, see Table 1 for details)

Table A7. Comparison between TGS sensors included on the low-cost logging system.

| Type | Target gas | Approximate price | Comments |
|---|---|---|---|
| 2600 | $C_2H_5OH$, $C_4H_{10}$, CO, $H_2$, $CH_4$ | 15 \$us | Designed as a smoke detector. |
| 2611-C00 | $CH_4$, $C_2H_5OH$, $C_4H_{10}$, CO, $H_2$, | 20 \$us | Designed for $CH_4$ detection. Fast response. |
| 2611-E00 | $CH_4$, $H_2$ | 20 \$us | Designed for $CH_4$ detection. Increased selectivity due to a carbon filter installed on top of the sensing material. |

25. **RC2: L101-103: Does weather during the sampling time frame have any impact on subsequent analysis? For example, Figure A4 shows the humidity and temperature ranges during the experiment. During the summer months with substantially higher water vapor mixing ratios, do the H2O vapor influences ever become nonlinear? As an aside, what causes the temperature spikes in Fig A4? Is this heating from the MOS?**

For each experiment we characterize the effect of $H_2O$ independently by selecting observations of ambient air without the presence of artificial spikes. Then we fit a linear model between $H_2O$ and voltage measurements from the TGS sensor to determine the sensitivities to $H_2O$. The figure below shows the fit of the linear models for each experiment. We have employed a linear relationship to derive the sensitivities to $H_2O$ on both experiments obtaining a reasonable correction of $H_2O$ effect on the baseline of TGS. While there is possibly a non-linear relationship on periods with high humidity, our experiment doesn't allows us to have such conclusion due to our limited dataset on the second experiment (only 1 month of data at the end of summer).

[Figure]

Sensitivities of TGS voltage to $H_2O$ mole fraction. The left plot shows the sensitivities for the first experiment (October 2019 – March 2020), and the right plot for the second experiment (August 2020 – September 2020). The red line is the fitted linear model.

The spikes observed in the temperature on Figure A4 are linked to fluctuations of temperature in the conditioned laboratory room. The magnitudes show the temperature inside the chamber which is affected by the heating from the MOS.

In summary, yes the periods with high humidity can produce non-linear behavior on the TGS baseline, but it is difficult to provide a general answer due to lack of available data on those periods (spring-summer).

26. **RC2: L128-129: Does the time constant differ for the TGS C and E sensors? Is this one reason for the higher phase mismatch when training the models with the Type E data?**

No, the time constant applied to the reference instrument is the same for type C and E sensors. The phase mismatch is linked to the carbon filter included on the Type E sensor.

We have added a sentence to clarify:

To determine the time constant (τ) of the buffer effect of the chambers, we applied an exponential weighted moving average (EWMA) to the CRDS data with different values of τ and compared them with the shape of the response of the TGS sensor (see Fig. A1). The time constant applied on the reference instrument to compare both TGS sensor types is the same.

27. **RC2: L144-145: As in point 4, does the H2O-voltage relationship ever become nonlinear at high enough humidity?**

    Commentary already addressed on point 4.

28. **RC2: L224-225: Will the spikes that end up being influential have any dependence on the structure in the data? In other words, is this method applicable when your dominant spike structure for a real sampling site may be unknown ahead of time?**

    Yes, in our methodology the typical clusters to which the sensors would be exposed need to be known beforehand. In the case of a real sampling site, the data used to train the models need to have examples of the typical shapes and magnitudes of spikes, therefore short sampling periods of the sensors collocated with a reference instrument would be required.

    A statement was added in the discussion section:

    This approach is designed with an a priori knowledge of the typical concentrations the sensors will be exposed. Although, exposing to a wide range of concentrations, like the ones included on cluster 9 from our experiment, can lead to have a large variety of examples for the training of the calibration models.

29. **RC2: L309-311: Why does the interquartile range increase when more training data is used?**

    We attribute this behavior to two problems, the first is probably due to redundancy in the training set that condition the model coefficients to certain values and thus producing higher error on validation sets that differ much from the more common values present in the training set. The second, could be linked to the presence of few spikes on each test set of the 20-fold cross validation making that the diversity of phase errors are more represented in the summary statistics.

30. **RC2: L315-317: I think this is an important result from a policy perspective, where monitoring emission magnitudes may still be of value, even if the spike phase is not exact.**

    The reviewer is right with this remark. We have added a statement on the discussion section:

Models have produced a reasonable estimation of the magnitude, which is important from a policy perspective, since information of the magnitude can be of value when monitoring emission magnitudes despite the errors in reconstructing the phase of the peaks.

31. **RC2: L352-352: This sentence is unclear. The MSD is larger for Case 1 than Case 11.**

The reviewer is right with this remark. We were comparing Case 11 and 10, the typo was corrected. Below the corrected sentence:

First, the smallest error did not correspond to the most parsimonious training set (Case 11) but to a larger training set (Case 10, 25% of the data).

32. **RC2: L383-385: This result speaks to point 7 above. If your peak clustering in your training data set differs (by some threshold) from your observational set, the parsimonious training often fails to meet the acceptable RMSE. Does this imply that you would need to bring a CRDS instrument to the field site for short-term sampling? It would be worth clarifying an optimal strategy for field deployment/field calibration.**

As mentioned in the answer on point 7, periodical short calibrations would be needed in order to update the clusters, and to achieve that the sensors would need to be collocated with a high precision instrument. Concerning the optimal strategy for field deployment, from the results of the ageing effect on the sensors, if the clustering structure is invariable, or has low variation, the models would not need to be trained for long periods of time, after six months the models achieve to meet the target requirement. If they are deployed on a site with high variability on the clustering structure, periodic re-calibrations would be needed.

We have added two sentences on the discussion section:

These result shows that models would require less calibrations in environments with low variations, or invariability, on the clustering structure. But for a deployment on sites with high variability on the clustering structure, periodic re-calibrations would be needed.

33. **RC2: Figure 1: Make image a larger so chamber set up is more easily distinguished.**

Figure updated.

[Figure]

Figure 1. (a) Example of a chamber with three sensors inside, (b) Scheme of the spike creation experiment.

34. **RC2: Figure 1: Is chamber D mentioned in the text anywhere?**

    The diagram was corrected. Chamber D was assembled but due to logging errors the data was not usable, so it was removed from the study.

    We have renamed all the chambers from A to E in the manuscript to prevent further confusion.

35. **RC2: Figure 4: Mislabeled as Type E, not Type C**

    Label on figure corrected.

TGS 2611-C00

[Figure]

**Figure 4. Example of reconstruction of the CRDS reference CH₄ signal on a time step of 5 s for a few spikes in the test set by (a) a Linear model, (b) a polynomial model, (c) a Random Forest model, (d) a Random Forest Hybrid model and (e) a Multi-Layer Perceptron model trained with 70% of data and using as input data from the TGS 2611-C00 sensors only.**

The right panels show scatter plots between the reference CH$_4$ signal and the modelled outputs. The colour code is the density of observations.

36. **RC2: Figure 9: It may be helpful to label the input data on the panels directly for ease of interpretation, if it's not too wordy.**
We decided to keep the figure as it, since the inclusion of the input data would take a large place and probably would lead to more confusion.